# Rapid $O_3$ assimilations – Part 2: tropospheric $O_3$ changes accompanied by declines in $NO_x$ emissions in the US and Europe in 2005-2020

Rui Zhu[1], Zhaojun Tang[1], Xiaokang Chen[1], Xiong Liu[2] and Zhe Jiang[1]*

[1]School of Earth and Space Sciences, University of Science and Technology of China, Hefei, Anhui, 230026, China.
[2]Center for Astrophysics | Harvard & Smithsonian, Cambridge, MA 02138, USA.

*Correspondence to: Zhe Jiang (zhejiang@ustc.edu.cn)

## Abstract

Tropospheric nitrogen dioxide ($NO_2$) concentrations have declined dramatically over the United States (US) and Europe in recent decades. Here we investigate the changes in surface and free tropospheric $O_3$ accompanied by $NO_2$ changes over the US and Europe in 2005-2020 by assimilating the Ozone Monitoring Instrument (OMI), and US Air Quality System (AQS) and European AirBase network $O_3$ observations. The assimilated $O_3$ concentrations demonstrate good agreement with $O_3$ observations: surface $O_3$ concentrations are 41.4, 39.5 and 39.5 ppb (US) and 35.3, 32.0 and 31.6 ppb (Europe); and tropospheric $O_3$ columns are 35.5, 37.0 and 36.8 DU (US) and 32.8, 35.3 and 36.4 DU (Europe) in the simulations, assimilations and observations, respectively. We find overestimated summertime surface $O_3$ concentrations over the US and Europe, which resulted in a surface $O_3$ maximum in July-August in simulations in contrast to April in observations. Furthermore, our analysis exhibits limited changes in surface $O_3$ concentrations, i.e., decreased by -6% over the US and increased by 1.5% over Europe in 2005-2020. The surface observation-based assimilations suggest insignificant changes in tropospheric $O_3$ columns: -3.0% (US) and 1.5% (Europe) in 2005-2020. While the OMI-based assimilations exhibit larger decreases in tropospheric $O_3$ columns, -12.0% (US) and -15.0% (Europe) in 2005-2020, the decreases mainly occurred in 2010-2014, corresponding to the reported slowed declines in free tropospheric $NO_2$ since 2010. Our analysis thus suggests limited impacts of local emission declines on tropospheric $O_3$ over the

US and Europe and advises more efforts to evaluate the possible contributions of natural
sources and transport. The discrepancy in assimilated tropospheric $O_3$ columns further
indicates the possible uncertainties in the derived tropospheric $O_3$ changes.

## 1. Introduction

The successful emission regulations employed in the United States (US) and Europe
(Crippa et al., 2016; EPA, 2017) have led to dramatic decreases in anthropogenic $NO_x$
emissions (Di et al., 2020; Macdonald et al., 2021; Jiang et al., 2022). As an important air
pollutant, tropospheric ozone ($O_3$) is produced when volatile organic compounds (VOCs) are
photochemically oxidized in the presence of nitrogen oxides ($NO_x$). As a major precursor of
tropospheric $O_3$, decreases in surface nitrogen dioxide ($NO_2$) concentrations, driven by declines
in $NO_x$ emissions, have led to marked decreases in surface $O_3$ concentrations over the US and
Europe in recent decades. For example, Chen et al. (2021) found a decrease in surface $O_3$
concentrations from approximately 60 to 45 ppb over the US in 1990-2019; Seltzer et al. (2020)
exhibited a decreasing trend of surface $O_3$ by approximately 0.8 ppb $yr^{-1}$ over the US in 2000-
2015; and Yan et al. (2018) found a decreasing trend of surface $O_3$ concentrations by
approximately 0.32 $\mu g/m^3/y$ over Europe in 1995-2014.
While $NO_x$ emissions are declining, the shift of $NO_x$ sources from power generation to
industrial and transportation sectors has led to diminishing effects on $NO_x$ emission controls
(Jiang et al., 2022). Furthermore, recent studies have demonstrated a slowdown in tropospheric
$NO_2$ column declines with respect to surface $NO_2$ concentrations over the US since
approximately 2010 (Jiang et al., 2018; Laughner and Cohen, 2019; Qu et al., 2021). Jiang et
al. (2022) further indicated a slowdown of declines in tropospheric $NO_2$ columns with respect
to surface $NO_2$ concentrations over both the US and Europe. Unlike surface $O_3$, which is
strongly affected by local emissions, free tropospheric $O_3$ is more susceptible to the influences
of free tropospheric sources and sinks, long-range transport, and stratospheric intrusion (Jiang
et al., 2015; Xue et al., 2021; Trickl et al., 2020). The different trends in surface and free
tropospheric $NO_2$ may thus result in different changes in surface and free tropospheric $O_3$ over
the US and Europe.

A single $O_3$ tracer mode (tagged-$O_3$) of the GEOS-Chem model was developed in the

companion paper (Zhu et al., 2023) and was combined with Ozone Monitoring Instrument
(OMI) and surface $O_3$ observations in China in 2015-2020 via a sequential Kalman filter (KF)
assimilation system (Tang et al., 2022; Han et al., 2022). The rapid $O_3$ assimilation capability
with approximately 91-94% reductions in computational costs (Zhu et al., 2023) provides a
new opportunity to extend atmospheric $O_3$ observations and mitigate the influence of
uncertainties in physical and chemical processes (Li et al., 2019; Chen et al., 2022) and
emission inventories (Zheng et al., 2017; Jiang et al., 2022). As the second part of this work,
we assimilate OMI and US Air Quality System (AQS) and European AirBase network $O_3$
observations in this work to constrain tropospheric $O_3$ in the US and Europe in 2005-2020 with
a $0.5° \times 0.625°$ horizontal resolution. A comparative analysis by assimilating satellite and
surface $O_3$ observations is useful for better characterization of $O_3$ changes in the surface and
free troposphere. Furthermore, this analysis helps evaluate the long-term performance of the
GEOS-Chem model in simulating tropospheric $O_3$ and can provide new insights into
tropospheric $O_3$ changes accompanied by the reported changes in tropospheric $NO_2$.

This paper is organized as follows: in Section 2, we provide descriptions for the AQS,

AirBase and OMI $O_3$ observations and the single $O_3$ tracer simulation and assimilation system
used in this work. We refer the reader to the companion paper (Zhu et al., 2023) for more details
about the atmospheric $O_3$ observations and the development and performance of the single $O_3$
tracer assimilation system. Tropospheric $O_3$ changes in the US and Europe in 2005-2020 are
demonstrated in Section 3 by assimilating atmospheric $O_3$ observations. As shown in Fig. 1,
five regions (i.e., Great Lakes (#1), Northeast US (#2), West Coast (#3), Middle US (#4) and
Southeast US (#5)) are defined within the US domain, and five regions (i.e., Britain (#1),
Central EU (#2), Western EU (#3), Iberian Peninsula (#4) and Apennine Peninsula (#5)) are
defined within the European domain based on anthropogenic $NO_x$ emissions in 2015. Regions
#1-3 (US) and regions #1-2 (Europe) are defined as highly polluted regions by excluding grids
with low and medium anthropogenic $NO_x$ emissions. Tropospheric $O_3$ changes over these
regions will be discussed to investigate the possible regional discrepancies in surface and free
tropospheric $O_3$ changes associated with different local pollution levels. Our conclusions
follow in Section 4.

## 2. Data and Methods

### 2.1 OMI and surface $O_3$ measurements

The OMI $O_3$ profile retrieval product (PROFOZ v0.9.3, level 2, Liu et al., 2010; Huang
et al., 2017) from the Smithsonian Astrophysical Observatory (SAO) was assimilated in this
work. The OMI instrument provides global covered measurements with backscattered sunlight
in the ultraviolet–visible range from 270 to 500 nm (UV1: 270–310 nm; UV2: 310–365 nm;
visible: 350–500 nm) with a spatial resolution of 13 × 24 km (nadir view). Following Huang et
al. (2017), the following filters are applied: 1) nearly clear-sky scenes with effective cloud
fraction < 0.3; 2) solar zenith angles (SZA) < 75°; and 3) fitting root mean square (RMS, ratio
of fitting residuals to assumed measurement error) < 2.0. Starting in 2009, anomalies were
found in OMI data and diagnosed as attenuated measured radiances in certain cross-track
positions. This instrument degradation has been referred to as the "row anomaly". To enhance
the quality and stability of data, only across-track positions between 4-11 (within 30 positions
in the UV1 channels) are used in our analysis. This treatment is similar to the production of
row-isolated data by using across-track positions between 3-18 (within 60 positions in the UV2
channels) in the OMI/MLS $O_3$ data (Ziemke et al., 2019; Wang et al., 2022).
We use in situ hourly surface $O_3$ measurements from the US AQS and European
Environment Agency AirBase networks. The AQS and AirBase networks collect ambient air
pollution data from monitoring stations located in urban, suburban, and rural areas. To ensure
the long-term stability of the observation record, we only considered stations with at least 14
years of observation records in 2005-2020. Observations provided by the AQS and AirBase
stations have been widely used in previous studies to investigate the sources and variabilities
of surface $O_3$ pollution (Shen et al., 2015; Boleti et al., 2020; He et al., 2022).
**2.2 Single $O_3$ tracer simulation and assimilation system**
The GEOS-Chem chemical transport model (http://www.geos-chem.org, version 12-8-1)
is driven by assimilated meteorological data of MERRA-2. The GEOS-Chem full chemistry
simulation includes fully coupled $O_3$-$NO_x$-VOC-halogen-aerosol chemistry. Our analysis is
conducted at a horizontal resolution of nested $0.5° \times 0.625°$ over the US and Europe with
chemical boundary conditions archived every 3 hours from global simulations with $4° \times 5°$
resolution. Emissions are computed by the Harvard-NASA Emission Component (HEMCO).
Global default anthropogenic emissions are from the CEDS (Community Emissions Data
System) (Hoesly et al., 2018). Regional emissions are replaced by MEIC (Multiresolution
Emission Inventory for China) in China, MIX in other regions of Asia (Li et al., 2017) and
NEI2011 in the US. Open fire emissions are from the Global Fire Emissions Database (GFED4)
(van der Werf et al., 2010).
Following Jiang et al. (2022), the total anthropogenic $NO_x$ and VOC emissions in the
GEOS-Chem model are scaled with the corresponding bottom-up inventories (MEIC for China,
NEI2014 for the US and ECLIPSE for Europe) so that the modeled surface nitrogen dioxide
($NO_2$) and VOC concentrations in the a priori simulations are identical to Jiang et al. (2022) in
2005-2018. The total anthropogenic $NO_x$ and VOC emissions in 2019-2020 in China, the US
and Europe are further scaled based on linear projections. The total anthropogenic $NO_x$
emissions in the a priori simulations declined by 53% (US) and 50% (Europe) in 2005-2020.
The total anthropogenic VOC emissions in the a priori simulations declined by 19% (US) and
33% (Europe) in 2005-2020. We refer the reader to Jiang et al. (2022) for the details of the
model configuration and performance, particularly the modeled trends of surface and
tropospheric column $NO_2$ in 2005-2018.
A new single $O_3$ tracer mode (tagged-$O_3$) was developed in the companion paper (Zhu et
al., 2023) by reading the archived production (PO3) and loss (LO3) of $O_3$ provided by the full
chemistry simulation. The major advantage of the single $O_3$ tracer mode is dramatic reductions
in computational costs by approximately 91-94% (Zhu et al., 2023). Fig. S1 and Fig. S2 (see
the SI) show the annual and seasonal averages of surface maximum daily 8-
hour average (MDA8) $O_3$ over the US and Europe in 2005-2020 from the full chemistry and
single $O_3$ tracer simulations (i.e., the a priori simulations in this work), respectively. We find
good spatial (Fig. S1 and Fig. S2) as well as temporal (Fig. S3, see the SI) consistencies in
surface MDA8 $O_3$ between full chemistry and single $O_3$ tracer simulations over the US and
Europe in 2005-2020. The computation costs (hours of wall time for one year simulation) are
160.7 and 9.4 hours within the nested US domain (0.5°×0.625°) and 103.4 and 6 hours within
the nested Europe domain (0.5°×0.625°) by full chemistry and single $O_3$ tracer mode,
respectively.
The low computational costs of the single $O_3$ tracer mode allow us to perform $O_3$
assimilations more efficiently. The sequential KF was conducted to assimilate AQS, AirBase
and OMI $O_3$ observations to produce the a posteriori $O_3$ concentrations. As a brief description
of the assimilation algorithm, the forward model (**M**) predicts the $O_3$ concentration ($x_{at}$) at
time t:

$$x_{at} = M_t x_{t-1} \quad (Eq. 1)$$

The optimized $O_3$ concentrations can be expressed as:
$$x_t = x_{\text{at}} + \mathbf{G}_t(y_t - \mathbf{K}_t x_{\text{at}}) \quad (\text{Eq.}\,2)$$
where $y_t$ is the observation (i.e., OMI or surface $O_3$ observations) and $\mathbf{K}_t$ represents the
operation operator that projects $O_3$ concentrations from the model space to the observation
space. $\mathbf{G}_t$ is the KF gain matrix, which can be described as:
$$\mathbf{G}_t = \mathbf{S}_{\text{at}}\mathbf{K}_t^T(\mathbf{K}_t\mathbf{S}_{\text{at}}\mathbf{K}_t^T + \mathbf{S}_\epsilon)^{-1} \quad (\text{Eq.}\,3)$$
where $\mathbf{S}_{\text{at}}$ and $\mathbf{S}_\epsilon$ are the model and observation covariances, respectively. The modeled
tropospheric $O_3$ profiles in the OMI-based assimilation processes are convolved by using the
OMI retrieval averaging kernels. The mixing of $O_3$ precursors in the planetary boundary layer
is considered with a simplified planetary boundary layer parameterization in surface
observation-based assimilations. We refer the reader to the companion paper (Zhu et al., 2023)
for more details about the development and performance of the single $O_3$ tracer assimilation
system by assimilating satellite and surface $O_3$ observations.

## 3. Results and Discussion
### 3.1 Surface $O_3$ by assimilating surface $O_3$ observations
Fig. 2a-e and Fig. 3a-e show the annual and seasonal averages of surface MDA8 $O_3$
observations from US AQS and European AirBase stations in 2005-2020. Fig. 2k-o and Fig.
3k-o further show the annual and seasonal averages of the a posteriori $O_3$ concentrations by
assimilating AQS or AirBase $O_3$ observations. As shown in Fig. 4 and Fig. 5, the assimilated
$O_3$ concentrations (blue lines) show good agreements with surface $O_3$ observations (red lines):
the mean surface MDA8 $O_3$ in 2005-2020 are 41.4, 39.5 and 39.5 ppb (US), 40.0, 37.7 and
38.2 ppb (Great Lakes), 38.1, 36.4 and 37.4 ppb (Northeast US), 41.6, 41.2 and 41.0 ppb (West
Coast), 42.2, 40.4 and 39.7 ppb (Middle US), 44.4, 40.3 and 39.9 ppb (Southeast US) in the a
priori simulations, a posteriori simulations and AQS observations, respectively; the mean
surface MDA8 $O_3$ in 2005-2020 are 35.3, 32.0 and 31.6 ppb (Europe), 29.9, 26.0 and 24.4 ppb
(Britain), 30.5, 28.2 and 28.0 ppb (Central EU), 35.9, 32.5 and 32.3 ppb (Western EU), 40.3,
35.2 and 34.2 ppb (Iberian Peninsula), 41.8, 35.3 and 34.0 ppb (Apennine Peninsula) in the a
priori simulations, a posteriori simulations and AirBase observations, respectively.

Similar to China, we find overestimated summertime surface $O_3$ concentrations in the a

priori simulations over the US and Europe (Fig. 4 and Fig. 5). However, in contrast to the
underestimated $O_3$ declines in June-July in China (Zhu et al., 2023), the overestimated
summertime $O_3$ over the US and Europe are caused by overestimated increases in surface $O_3$
in July-August, which have led to surface MDA8 $O_3$ maximum in July-August in the
simulations. In contrast, assimilations suggest that surface $O_3$ is broadly maximum in April
over the US and Europe (Fig. 4 and Fig. 5), although $O_3$ seasonality varies over different
regions. We find good agreements in surface $O_3$ concentrations between a priori and a
posteriori simulations over the US in seasons outside of summer (Fig. 2p-t), in contrast to the
large differences between a priori and a posteriori simulations over Europe (Fig. 3p-t in this
work) and China (Zhu et al., 2023). The inaccurate surface $O_3$ concentrations over three
continents reveal possible uncertainties in model simulations, particularly the contributions
from natural and anthropogenic processes; for example, the higher temperature and solar
radiation can lead to high $O_3$ concentrations in August, whereas the transport of $O_3$ and its
precursors can lead to high $O_3$ concentrations in April (Parrish et al., 2013).

Furthermore, our analysis exhibits high surface MDA8 $O_3$ concentrations over the West

Coast (41.2 ppb) in the US. Except for the West Coast, the assimilated surface MDA8 $O_3$
concentrations are lower over areas with higher anthropogenic $NO_x$ emissions over the US and
Europe. For example, 37.7 and 36.4 ppb in the Great Lakes and Northeast US, respectively, in
contrast to 40.4 and 40.3 ppb in the Middle US and Southeast US, respectively; and 26.0 and
28.2 ppb in the Britain and Central EU, respectively, in contrast to 32.5, 35.2 and 35.3 ppb in

the Western EU, Iberian Peninsula and Apennine Peninsula, respectively. The inverse relationships between surface $O_3$ concentrations and local anthropogenic $NO_x$ emissions indicate the important impacts of natural sources and meteorological conditions on surface $O_3$ pollution over the US and Europe because of continuous declines in anthropogenic emissions in the past decades. This is the opposite of the higher $O_3$ concentrations in areas with higher local anthropogenic $NO_x$ emissions in China (Zhu et al., 2023), where surface $O_3$ pollution is strongly affected by anthropogenic emissions.

**3.2 Limited changes in surface $O_3$ concentrations**

Following Jiang et al. (2022), the anthropogenic $NO_x$ and VOC emissions over the US in 2005-2020 declined by 53% (-5.1% $yr^{-1}$) and 19% (-1.4% $yr^{-1}$) in our a priori simulations, which is accompanied by slight decreasing trends in surface MDA8 $O_3$ in the a priori simulations (Table 1.1): -0.29 (spring), -0.45 (summer), -0.07 (autumn) and 0.05 (winter) ppb $yr^{-1}$; and the relative trends are -0.7 (spring), -0.9 (summer), -0.2 (autumn) and 0.2 (winter) % $yr^{-1}$. Similarly, the anthropogenic $NO_x$ and VOC emissions over Europe in 2005-2020 declined by 50% (-4.4% $yr^{-1}$) and 33% (-2.7% $yr^{-1}$) in our a priori simulations, which is accompanied by slightly increasing trends of surface MDA8 $O_3$ in the a priori simulations (Table 2.1): -0.07 (spring), -0.07 (summer), 0.07 (autumn) and 0.24 (winter) ppb $yr^{-1}$; and the relative trends are -0.2 (spring), -0.2 (summer), 0.2 (autumn) and 1.0 (winter) % $yr^{-1}$. It is surprising to see the limited changes in surface $O_3$ concentrations in the simulations accompanied by dramatic declines in anthropogenic emissions.

We thus further investigate the changes in surface $O_3$ by assimilating surface $O_3$ observations. As shown in Table 1.1 and Fig. 6k-o, our assimilations suggest -0.27 (spring), -0.46 (summer), -0.12 (autumn) and 0.11 (winter) ppb $yr^{-1}$ changes in surface MDA8 $O_3$ over the US in 2005-2020, and the relative changes are -0.6 (spring), -1.0 (summer), -0.3 (autumn) and 0.4 (winter) % $yr^{-1}$. Similarly, as shown in Table 2.1 and Fig. 7k-o, our assimilations

suggest -0.04 (spring), -0.03 (summer), 0.09 (autumn) and 0.19 (winter) ppb $yr^{-1}$ changes in
surface MDA8 $O_3$ over Europe in 2005-2020, and the relative changes are -0.1 (spring), -0.1
(summer), 0.3 (autumn) and 0.9 (winter) % $yr^{-1}$. In contrast to the underestimated increasing
trends in surface $O_3$ concentrations in the a priori simulations in China (Zhu et al., 2023), we
find broadly consistent trends between simulations and assimilations over the US and Europe,
which confirms the limited changes in surface $O_3$ concentrations over the US and Europe.
The changes in surface $O_3$ concentrations have marked regional and seasonal
discrepancies. As shown in Tables S1-S5 (see the SI), our assimilations demonstrate stronger
increasing trends in surface $O_3$ concentrations in 2005-2020 in the winter (0.39 ppb $yr^{-1}$ or 1.5%
$yr^{-1}$) over the Great Lakes, in the winter (0.36 ppb $yr^{-1}$ or 1.4% $yr^{-1}$) over the Northeast US, in
the autumn (0.34 ppb $yr^{-1}$ or 0.8% $yr^{-1}$) and winter (0.29 ppb $yr^{-1}$ or 1.0% $yr^{-1}$) over the West
Coast, as well as decreasing trends in surface $O_3$ concentrations in 2005-2020 in the summer
over the Great Lakes (-0.51 ppb $yr^{-1}$ or -1.0% $yr^{-1}$), Northeast US (-0.52 ppb $yr^{-1}$ or -1.1% $yr^{-1}$),
Middle US (-0.61 ppb $yr^{-1}$ or -1.3% $yr^{-1}$) and Southeast US (-0.87 ppb $yr^{-1}$ or -1.9% $yr^{-1}$). The
areas with higher anthropogenic $NO_x$ emissions such as the Great Lakes and Northeast US
demonstrate lower surface $O_3$ concentrations and are accompanied by stronger increasing
trends in the winter and weaker decreasing trends in the summer.
Tables S6-S10 (see the SI) further show the details of tropospheric $O_3$ changes in Europe.
Our assimilations demonstrate stronger increasing trends in surface $O_3$ concentrations in 2005-
2020 in the winter over Britain (0.28 ppb $yr^{-1}$ or 1.5% $yr^{-1}$), Central EU (0.26 ppb $yr^{-1}$ or 1.5%
$yr^{-1}$), Western EU (0.25 ppb $yr^{-1}$ or 1.1% $yr^{-1}$), Iberian Peninsula (0.17 ppb $yr^{-1}$ or 0.6% $yr^{-1}$)
and Apennine Peninsula (0.18 ppb $yr^{-1}$ or 0.8% $yr^{-1}$), as well as decreasing trends in surface $O_3$
concentrations in 2005-2020 in the summer (-0.07 ppb $yr^{-1}$ or -0.2% $yr^{-1}$) over Britain, in the
summer (-0.10 ppb $yr^{-1}$ or -0.2% $yr^{-1}$) over the Western EU, in the summer (-0.20 ppb $yr^{-1}$ or -
0.5% $yr^{-1}$) over the Iberian Peninsula and in the spring (-0.09 ppb $yr^{-1}$ or -0.2% $yr^{-1}$) over the
Apennine Peninsula. Similar to the US, areas with higher anthropogenic $NO_x$ emissions such
as Britain and Central EU demonstrate lower surface $O_3$ concentrations and are accompanied
by stronger increasing trends in the winter and weaker decreasing trends in the summer.

Furthermore, Zhu et al. (2023) demonstrated a large discrepancy in the trends in

assimilated surface $O_3$ between urban (i.e., areas with air quality stations) and regional
backgrounds in China in 2015-2020: 3.0% $yr^{-1}$ (sampled at air quality stations) and 1.3% $yr^{-1}$
(land average). In contrast, we did not find a comparable discrepancy over the US and Europe:
the trends of assimilated surface $O_3$ are -0.4% $yr^{-1}$ (Table 1.1, sampled at AQS $O_3$ observations)
and -0.4% $yr^{-1}$ (Table 1.2, land average) over the US and -0.2% $yr^{-1}$ (Table 2.1, sampled at
AirBase $O_3$ observations) and 0.0% $yr^{-1}$ (Table 2.2, land average) over Europe. The difference
between China and the US/Europe suggests more consistent changes in surface $O_3$ between
urban and regional background areas in the US and Europe. This implies possible larger relative
contributions of regional background $O_3$ to surface $O_3$ observations in the US and Europe,
which could be associated with the limited changes in surface $O_3$ concentrations in 2005-2020
because regional background $O_3$ is less sensitive to changes in anthropogenic $NO_x$ and VOC
emissions.
**3.3 Tropospheric $O_3$ columns by assimilating OMI $O_3$ observations**

Fig. S4a-e and Fig. S5a-e (see the SI) show the annual and seasonal averages of

tropospheric OMI $O_3$ columns in 2005-2020 over the US and Europe, respectively. Fig. S4k-o
and Fig. S5k-o further show the annual and seasonal averages of the a posteriori tropospheric
$O_3$ columns by assimilating OMI $O_3$ observations. The assimilated tropospheric $O_3$ columns
show good agreement with OMI $O_3$ observations: the mean tropospheric $O_3$ columns over the
US in 2005-2020 (Table 1.3) are 35.5 DU in the a priori simulations, and 37.0 and 36.8 DU in
the a posteriori simulations and OMI observations, respectively; the mean tropospheric $O_3$
columns over Europe in 2005-2020 (Table 2.3) are 32.8 DU in the a priori simulations, and
35.3 and 36.4 DU in the a posteriori simulations and OMI observations, respectively. However,
there are small deviations in the trends between assimilations and OMI observations. As shown
in Fig. S6-S7 (see the SI), the trends of tropospheric $O_3$ columns over the US in 2005-2020
(Table 1.3) are -0.11 DU $yr^{-1}$ in the a priori simulations, and -0.16 and -0.01 DU $yr^{-1}$ in the a
posteriori simulations and OMI observations, respectively; the trends of tropospheric $O_3$
columns over Europe in 2005-2020 (Table 2.3) are -0.09 DU $yr^{-1}$ in the a priori simulations,
and -0.25 and -0.15 DU $yr^{-1}$ in the a posteriori simulations and OMI observations, respectively.
These deviations are associated with the adjustments to regional $O_3$ boundary conditions in the
nested assimilations by assimilating global OMI $O_3$ observations, reflecting the different
changes in OMI $O_3$ between US/Europe continents and global backgrounds. For example, the
mean tropospheric $O_3$ columns over the US in 2005 are 36.5 DU in OMI observations, and 35.9
and 37.5 DU in the assimilations by reading a priori and adjusted $O_3$ boundary conditions,
respectively; the mean tropospheric $O_3$ columns over Europe in 2005 are 37.5 DU in OMI
observations, and 34.6 and 36.9 DU in the assimilations by reading a priori and adjusted $O_3$
boundary conditions, respectively.

The annual averages of surface MDA8 $O_3$ in the a priori simulation and assimilations are

35.3 and 32.0 ppb with a relative difference of 10% over Europe (Table 2.1); 41.4 and 39.5 ppb
with a relative difference of 5% over the US (Table 1.1); and 42.9 and 41.8 ppb with a relative
difference of 3% over China (Zhu et al., 2023). In addition, the annual averages of tropospheric
$O_3$ columns in the a priori simulation and assimilations are 32.8 and 35.3 DU with a relative
difference of -7% over Europe (Table 2.3); 35.5 and 37.0 DU with a relative difference of -4%
over the US (Table 1.3); and 37.1 and 37.9 DU with a relative difference of -2% over China
(Zhu et al., 2023). It seems that the GEOS-Chem model has a better performance in regional
averages of surface and free tropospheric $O_3$ concentrations in China and the US than in
Europe.
The output $O_3$ profiles from a priori and a posteriori simulations are convolved with OMI
averaging kernels in Fig. S4-S7. However, the convolution of OMI $O_3$ averaging kernels on
the output $O_3$ profiles can affect the weights of the derived tropospheric columns to $O_3$ at
different vertical levels and thus may not accurately represent the actual tropospheric $O_3$
columns. Fig. 8 and Fig. 9 further show tropospheric $O_3$ columns from a priori and a posteriori
simulations, in which the output $O_3$ profiles are not convolved with OMI averaging kernels.
The assimilated tropospheric $O_3$ columns are 35.6 and 38.7 DU (US), 36.8 and 40.2 DU (Great
Lakes), 36.8 and 40.3 DU (Northeast US), 38.1 and 41.9 DU (West Coast), 38.9 and 41.5 DU
(Middle US), 43.5 and 45.8 DU (Southeast US) in 2005-2020 by assimilating AQS and OMI
$O_3$ observations, respectively; the assimilated tropospheric $O_3$ columns are 31.5 and 35.9 DU
(Europe), 29.7 and 34.7 DU (Britain), 30.4 and 34.9 DU (Central EU), 31.8 and 36.4 DU
(Western EU), 33.6 and 38.1 DU (Iberian Peninsula), 34.0 and 38.2 DU (Apennine Peninsula)
in 2005-2020 by assimilating AirBase and OMI $O_3$ observations, respectively. We find that
tropospheric $O_3$ columns obtained by assimilating surface $O_3$ observations are lower than those
obtained by assimilating OMI $O_3$ observations. Similar to surface $O_3$ concentrations,
tropospheric $O_3$ columns are lower over areas with higher anthropogenic $NO_x$ emissions over
the US and Europe such as the Great Lakes, Northeast US, Britain and Central EU. This is
opposite to the higher tropospheric $O_3$ columns over areas with higher local anthropogenic $NO_x$
emissions in China (Zhu et al., 2023).
In contrast to the surface MDA8 $O_3$ maximum in April in the observations (Fig. 4 and
Fig. 5), the assimilated tropospheric $O_3$ columns are broadly maximum in July-August over the
US and Europe (Fig. 10 and Fig. 11). The free tropospheric $O_3$ maximum in the summer has
been reported in previous studies. For example, Wespes et al. (2018) demonstrated a free
tropospheric $O_3$ maximum in summer over Europe by using Infrared Atmospheric Sounding
Interferometer (IASI) observations; Petetin et al. (2016) exhibited a free tropospheric $O_3$
maximum in summer over Europe by using MOZAIC aircraft measurements. We find good
agreement in the seasonality of free tropospheric $O_3$ between simulations and assimilations in
contrast to the inaccurate simulation of the seasonality of surface $O_3$ concentrations in the
simulations. More studies are needed in the future to explore the sources of this difference in
model performance.
Furthermore, Fig. S8-S9 (see the SI) demonstrate the $O_3$ vertical profiles in 2005-2009,
2010-2014 and 2015-2020, respectively. The assimilation of surface $O_3$ observations leads to
decreases in $O_3$ concentrations in the lower troposphere but has small impacts on free
tropospheric $O_3$. In contrast, the assimilation of OMI $O_3$ observations leads to dramatic
enhancements in $O_3$ concentrations in the middle and upper troposphere without noticeable
differences between areas with high and low local anthropogenic $NO_x$ emissions. The
enhancement in free tropospheric $O_3$ by assimilating OMI $O_3$ observations declined gradually
from 2005-2009 to 2015-2020. The adjustment in free tropospheric $O_3$ by assimilating OMI $O_3$
observations in 2015-2020 is larger but comparable with the adjustment in 2015-2020 in China
(Zhu et al., 2023).

**3.4 Large decreases in tropospheric $O_3$ columns**

Fig 12 and Fig. 13 show the trends in tropospheric $O_3$ columns in 2005-2020 from a priori
simulations and a posteriori simulations by assimilating surface and OMI $O_3$ observations. The
trends of tropospheric $O_3$ columns in 2005-2020 are -0.07, -0.07 and -0.29 DU yr$^{-1}$ (US), -0.03,
-0.03 and -0.29 DU yr$^{-1}$ (Great Lakes), -0.02, -0.02 and -0.31 DU yr$^{-1}$ (Northeast US), -0.02, -
0.01 and -0.26 DU yr$^{-1}$ (West Coast), -0.08, -0.07 and -0.24 DU yr$^{-1}$ (Middle US), -0.19, -0.18
and -0.28 DU yr$^{-1}$ (Southeast US) in the a priori simulations and a posteriori simulations by
assimilating AQS and OMI $O_3$ observations, respectively; and are 0.03, 0.03 and -0.36 DU yr$^{-}$
$^1$ (Europe), 0.00, 0.00 and -0.49 DU yr$^{-1}$ (Britain), 0.04, 0.04 and -0.38 DU yr$^{-1}$ (Central EU),
0.02, 0.03 and -0.36 DU yr$^{-1}$ (Western EU), 0.02, 0.02 and -0.30 DU yr$^{-1}$ (Iberian Peninsula), -
0.04, 0.04 and -0.26 DU yr$^{-1}$ (Apennine Peninsula) in the a priori simulations and a posteriori
simulations by assimilating AirBase and OMI O$_3$ observations, respectively. Our analysis thus
exhibits dramatically lower decreasing trends in tropospheric O$_3$ columns in the a priori
simulations and assimilations by assimilating surface O$_3$ observations with respect to OMI-
based assimilations.
The limited changes in surface O$_3$ concentrations in the a priori simulations and
assimilations by assimilating surface O$_3$ observations indicate limited influences of declines in
local anthropogenic emissions on surface O$_3$ concentrations in the US and Europe in 2005-
2020. We can thus expect insignificant influences of the vertical transport of surface O$_3$ on the
changes in free tropospheric O$_3$ over the US and Europe in 2005-2020, as illustrated by the flat
trends in tropospheric O$_3$ columns in the a priori simulations and assimilations by assimilating
surface O$_3$ observations (Fig. 10 and Fig. 11), as well as the small impacts of assimilation of
surface O$_3$ observations on free tropospheric O$_3$ (Fig. S8-S9). However, as indicated by Jiang
et al. (2022), tropospheric OMI NO$_2$ columns declined by 36% and 23% in 2005-2018 over the
US and Europe, respectively. Are the large decreases in tropospheric O$_3$ columns by
assimilating OMI O$_3$ observations, i.e., 12.0% (US) and 15.0% (Europe) in 2005-2020, caused
by the declines in free tropospheric NO$_2$?
As indicated by Jiang et al. (2022), tropospheric OMI NO$_2$ columns declined by -7.0%
yr$^{-1}$ (US) and -4.2% yr$^{-1}$ (Europe) in 2005-2010, which was followed by a dramatic slowdown
in the decreasing trends, i.e., -1.7% yr$^{-1}$ (US) and -1.2% yr$^{-1}$ (Europe) in 2010-2018. However,
as shown in Table 1.4, tropospheric O$_3$ columns obtained by assimilating OMI O$_3$ observations
declined by -0.3, -2.3 and -0.5% yr$^{-1}$ over the US in 2005-2009, 2010-2014 and 2015-2020,
respectively. Similarly, tropospheric O$_3$ columns obtained by assimilating OMI O$_3$
observations declined by -1.0, -2.3 and -0.8% yr$^{-1}$ over Europe (Table 2.4) in 2005-2009, 2010-
2014 and 2015-2020, respectively. The OMI-based declines in tropospheric O$_3$ columns over
the US and Europe mainly occurred in the period with slowed decreases in free tropospheric
$NO_2$ after 2010; in contrast, the dramatic declines in tropospheric $NO_2$ columns before 2010
were accompanied by limited changes in free tropospheric $O_3$. It is thus difficult to conclude
that the large decreases in tropospheric $O_3$ columns over the US and Europe in 2010-2014 are
dominated by declines in local anthropogenic $NO_x$ emissions.
We note our OMI-based analysis could be affected by the row anomaly issue, although
the usage of "row-isolated" data by using across-track positions between 4-11 in this work is
expected to reduce the impacts of row anomaly. As shown by Huang et al. (2017), the row
anomaly can lead to discontinuity in the trends in OMI $O_3$ observations in 2009. However, the
large decreases in tropospheric $O_3$ columns over the US and Europe mainly occurred after
2010. Consequently, we assume a limited influence of row anomaly on our conclusion.
Furthermore, OMI is sensitive to $O_3$ concentrations in the free troposphere; OMI-based
assimilations are driven by adjusted regional $O_3$ boundary conditions provided by global OMI
$O_3$ assimilations and can reflect optimized adjustments in both local and global background $O_3$
concentrations. In contrast, surface observations are sensitive to local $O_3$ concentrations;
surface observation-based assimilations are driven by the a priori $O_3$ boundary conditions,
which thus reflects the optimized adjustments in local contributions and is also affected by
lacking optimization on the impacts of $O_3$ precursors due to the single $O_3$ tracer simulations.
These factors contributed to the difference in the trends of tropospheric $O_3$ columns by
assimilating surface and satellite observations. Assimilations of both surface and satellite
observations, as shown in this work, are expected to provide more information to better
characterization of the changes and uncertainties in free tropospheric $O_3$.
**4. Conclusion**
As a companion paper of Zhu et al. (2023) which focuses on tropospheric $O_3$ change in
China in 2015-2020, this paper investigates the changes in surface and free tropospheric $O_3$
over the US and Europe in 2005-2020 by assimilating OMI, AQS and AirBase $O_3$ observations.
The assimilated $O_3$ concentrations demonstrate good agreement with $O_3$ observations: surface
$O_3$ concentrations are 41.4, 39.5 and 39.5 ppb (US) and 35.3, 32.0 and 31.6 ppb (Europe) in
the a priori and a posteriori simulations and AQS and AirBase $O_3$ observations, respectively;
and tropospheric $O_3$ columns are 35.5, 37.0 and 36.8 DU (US) and 32.8, 35.3 and 36.4 DU
(Europe) in the a priori and a posteriori simulations (convolved with OMI retrieval averaging
kernels) and OMI $O_3$ observations, respectively. The modeled surface $O_3$ by GEOS-Chem is
overestimated in the summer, which results in a surface $O_3$ maximum in July-August in the
simulations in contrast to April in the observations; in contrast, GEOS-Chem demonstrates
good performance in the simulation of seasonality in free tropospheric $O_3$, which is maximum
in July-August. In addition, we find lower surface $O_3$ concentrations over areas with higher
anthropogenic $NO_x$ emissions in the US and Europe. This is the opposite of the higher $O_3$
concentrations in areas with higher local anthropogenic $NO_x$ emissions in China (Zhu et al.,

2023).

Our analysis exhibits a noticeable decrease in surface $O_3$ concentrations over the US in

the summer by 15% in 2005-2020. However, accompanied by approximately 50% reductions
in $NO_x$ emissions, changes in surface $O_3$ concentrations are limited in Europe and other seasons
in the US: the annual surface MDA8 $O_3$ decreased by -6% over the US and increased by 1.5%
over Europe in 2005-2020, and the decreases in surface $O_3$ concentrations are weaker over
areas with higher local anthropogenic $NO_x$ emissions. Furthermore, the surface observation-
based assimilations suggest insignificant changes in tropospheric $O_3$ columns: -3.0% (US) and
1.5% (Europe) in 2005-2020. While the OMI-based assimilations exhibit large decreases in
tropospheric $O_3$ columns, i.e., -12.0% (US) and -15.0% (Europe) in 2005-2020, the decreases
in tropospheric $O_3$ columns mainly occurred in 2010-2014, corresponding to reported slowed
declines in free tropospheric $NO_2$ since 2010 (Jiang et al., 2022). Despite the dramatic declines
in tropospheric $NO_2$, particularly, declines in tropospheric $NO_2$ columns in 2005-2010, our
analysis suggests limited impacts of local emission declines on changes in tropospheric $O_3$ over
the US and Europe because the rapid decline in tropospheric $NO_2$ columns in 2005-2010
corresponds to relatively flat trends in tropospheric $O_3$. More efforts are suggested to evaluate
the contributions of natural sources and transport to tropospheric $O_3$ changes, which is critical
for making more effective policies to reduce $O_3$ pollution.

**Code and data availability:** The AQS and AirBase surface $O_3$ data can be downloaded from
https://www.eea.europa.eu/data-and-maps/data/aqereporting-8 and
https://aqs.epa.gov/aqsweb/airdata/download_files.html#Row. The OMI PROFOZ product
can be acquired at
https://avdc.gsfc.nasa.gov/pub/data/satellite/Aura/OMI/V03/L2/OMPROFOZ/. The GEOS-
Chem model (version 12.8.1) can be downloaded from http://wiki.seas.harvard.edu/geos-
chem/index.php/GEOS-Chem_12#12.8.1. The KPP module for tagged-$O_3$ simulations can be
downloaded from https://doi.org/10.5281/zenodo.7545944.

**Competing interests**: The contact author has declared that neither they nor their co-authors
have any competing interests.

**Acknowledgments:** We thank United States Environmental Protection Agency and the
European Environmental Agency for providing the surface $O_3$ measurements. The numerical
calculations in this paper have been done on the supercomputing system in the Supercomputing
Center of University of Science and Technology of China. This work was supported by the
Hundred Talents Program of Chinese Academy of Science and National Natural Science
Foundation of China (42277082, 41721002).

## Table and Figures

**Table 1.** Averages (with units ppb or DU) and trends (with units ppb yr$^{-1}$ or DU yr$^{-1}$) of surface and tropospheric column $O_3$ concentrations in 2005-2020 over the US from observations (AQS and OMI) and a priori and a posteriori (KF) simulations. T1.1): the modeled surface $O_3$ is sampled at the locations and times of AQS surface $O_3$ observations; T1.2): the modeled surface $O_3$ is averaged over the US (land only); T1.3): the output $O_3$ profiles from the a priori and a posteriori simulations are convolved with OMI $O_3$ averaging kernels; T1.4): the output $O_3$ profiles are NOT convolved with OMI $O_3$ averaging kernels. The uncertainties in the averages are calculated using the bootstrapping method. The trends and uncertainties in the trends are calculated using the linear fitting of averages by using the least squares method (see details in the SI).

**Table 2.** Averages (with units ppb or DU) and trends (with units ppb yr$^{-1}$ or DU yr$^{-1}$) of surface and tropospheric column $O_3$ concentrations in 2005-2020 over Europe from observations (AirBase and OMI) and a priori and a posteriori (KF) simulations. T2.1): the modeled surface $O_3$ are sampled at the locations and times of AirBase surface $O_3$ observations; T2.2): the modeled surface $O_3$ are averaged over Europe (land only); T2.3): the output $O_3$ profiles from the a priori and a posteriori simulations are convolved with OMI $O_3$ averaging kernels; T2.4): the output $O_3$ profiles are NOT convolved with OMI $O_3$ averaging kernels.

**Fig. 1.** (a) Anthropogenic $NO_x$ emissions over the US in 2015; (b) Region definitions for Great Lakes (#1), Northeast US (#2), West Coast (#3), Middle US (#4) and Southeast US (#5). Regions #1-3 are defined as highly polluted (HP) regions by excluding grids with low and medium anthropogenic $NO_x$ emissions. (c) Anthropogenic $NO_x$ emissions over Europe in 2015; (d) Region definitions for Britain (#1), Central EU (#2), Western EU (#3), Iberian Peninsula (#4) and Apennine Peninsula (#5). Regions #1 and #2 are defined as highly polluted (HP) regions by excluding grids with low and medium anthropogenic $NO_x$ emissions. The different colors (red, gray and green) represent grids with high (highest 15%), medium (15-50%) and low (lowest 50%) anthropogenic $NO_x$ emissions.

**Fig. 2.** Surface MDA8 $O_3$ over the US in 2005-2020 (annual and seasonal averages) from (a-e) AQS stations; (f-j) GEOS-Chem a priori simulation; (k-o) GEOS-Chem a posteriori

simulation by assimilating AQS $O_3$ observations. (p-t) Bias in the a priori simulations calculated by a priori minus a posteriori $O_3$ concentrations.

**Fig. 3.** Surface MDA8 $O_3$ over Europe in 2005-2020 (annual and seasonal averages) from (a-e) AirBase stations; (f-j) GEOS-Chem a priori simulation; (k-o) GEOS-Chem a posteriori simulation by assimilating AirBase $O_3$ observations. (p-t) Bias in the a priori simulations calculated by a priori minus a posteriori $O_3$ concentrations.

**Fig. 4.** (a-f) Daily averages of surface MDA8 $O_3$ over the US in 2005-2020 from AQS stations (red) and GEOS-Chem a priori (black) and a posteriori (blue) simulations by assimilating AQS $O_3$ observations. (g-l) Monthly averages of MDA8 $O_3$. The dashed lines in panels g-l are annual averages.

**Fig. 5.** (a-f) Daily averages of surface MDA8 $O_3$ over Europe in 2005-2020 from AirBase stations (red) and GEOS-Chem a priori (black) and a posteriori (blue) simulations by assimilating AirBase $O_3$ observations. (g-l) Monthly averages of MDA8 $O_3$. The dashed lines in panels g-l are annual averages.

**Fig. 6.** Trends of surface MDA8 $O_3$ over the US in 2005-2020 (annual and seasonal averages) from (a-e) AQS stations; (f-j) GEOS-Chem a priori simulation; (k-o) GEOS-Chem a posteriori simulation by assimilating AQS $O_3$ observations.

**Fig. 7.** Trends of surface MDA8 $O_3$ over Europe in 2005-2020 (annual and seasonal averages) from (a-e) AirBase stations; (f-j) GEOS-Chem a priori simulation; (k-o) GEOS-Chem a posteriori simulation by assimilating AirBase $O_3$ observations.

**Fig. 8.** Tropospheric $O_3$ columns over the US in 2005-2020 (annual and seasonal averages) from (a-e) GEOS-Chem a priori simulation; (f-j) Assimilations of AQS surface $O_3$ observations; (k-o) Assimilations of OMI $O_3$ observations. (p-t) Difference in tropospheric $O_3$ columns calculated by OMI-based assimilations minus surface observation-based assimilations.

**Fig. 9.** Tropospheric $O_3$ columns over Europe in 2005-2020 (annual and seasonal averages) from (a-e) GEOS-Chem a priori simulation; (f-j) Assimilations of AirBase surface $O_3$

observations; (k-o) Assimilations of OMI $O_3$ observations. (p-t) Difference in tropospheric $O_3$ columns calculated by OMI-based assimilations minus surface observation-based assimilations.

**Fig. 10.** (a-f) Daily averages of tropospheric $O_3$ columns over the US in 2005-2020 from GEOS-Chem a priori simulation (black) and a posteriori simulations by assimilating AQS (blue) and OMI (red) $O_3$ observations. (g-l) Monthly averages of tropospheric $O_3$ columns. The dashed lines in panels g-l are annual averages.

**Fig. 11.** (a-f) Daily averages of tropospheric $O_3$ columns over Europe in 2005-2020 from GEOS-Chem a priori simulation (black) and a posteriori simulations by assimilating AirBase (blue) and OMI (red) $O_3$ observations. (g-l) Monthly averages of tropospheric $O_3$ columns. The dashed lines in panels g-l are annual averages.

**Fig. 12.** Trends of tropospheric $O_3$ columns over the US in 2005-2020 (annual and seasonal averages) from (a-e) GEOS-Chem a priori simulation; (f-j) Assimilations of AQS surface $O_3$ observations; (k-o) Assimilations of OMI $O_3$ observations.

**Fig. 13.** Trends of tropospheric $O_3$ columns over Europe in 2005-2020 (annual and seasonal averages) from (a-e) GEOS-Chem a priori simulation; (f-j) Assimilations of AirBase surface $O_3$ observations; (k-o) Assimilations of OMI $O_3$ observations.

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

| United States | | | Annual | | Spring | | Summer | | Autumn | | Winter | |
|---|---|---|---|---|---|---|---|---|---|---|---|---|
| | | | Mean | Trend | Mean | Trend | Mean | Trend | Mean | Trend | Mean | Trend |
| T1.1 surface (sampled) | 2005-2020 | AQS | 39.5±0.2 | -0.18±0.04 | 45.4±0.2 | -0.26±0.06 | 45.2±0.3 | -0.49±0.10 | 36.2±0.2 | -0.18±0.09 | 31.5±0.3 | 0.14±0.05 |
| | | a priori | 41.4±0.2 | -0.18±0.04 | 44.2±0.1 | -0.29±0.04 | 51.2±0.3 | -0.45±0.11 | 39.2±0.2 | -0.07±0.06 | 30.9±0.2 | 0.05±0.05 |
| | | KF-AQS | 39.5±0.2 | -0.17±0.04 | 44.8±0.1 | -0.27±0.05 | 46.0±0.3 | -0.46±0.10 | 36.3±0.2 | -0.12±0.07 | 31.1±0.2 | 0.11±0.04 |
| T1.2 surface | 2005-2020 | a priori | 40.3±0.1 | -0.17±0.04 | 43.3±0.1 | -0.28±0.05 | 49.1±0.1 | -0.36±0.10 | 38.1±0.1 | -0.10±0.05 | 30.8±0.1 | 0.04±0.05 |
| | | KF-AQS | 39.2±0.1 | -0.15±0.03 | 43.5±0.1 | -0.25±0.04 | 46.1±0.1 | -0.34±0.09 | 36.4±0.1 | -0.12±0.05 | 31.0±0.1 | 0.07±0.04 |
| T1.3 trop. column (convolved) | 2005-2020 | OMI | 36.8±0.1 | -0.01±0.05 | 38.5±0.1 | 0.00±0.09 | 42.1±0.1 | 0.11±0.08 | 34.3±0.1 | -0.14±0.05 | 32.0±0.1 | -0.03±0.10 |
| | | a priori | 35.5±0.1 | -0.11±0.03 | 36.9±0.1 | -0.14±0.07 | 41.9±0.1 | -0.15±0.06 | 33.5±0.1 | -0.08±0.03 | 29.8±0.1 | -0.06±0.04 |
| | | KF-OMI | 37.0±0.1 | -0.16±0.04 | 39.4±0.1 | -0.21±0.07 | 43.3±0.1 | -0.02±0.06 | 34.6±0.1 | -0.18±0.04 | 30.7±0.1 | -0.21±0.04 |
| T1.4 trop. Column | 2005-2020 | a priori | 35.9±0.1 | -0.07±0.04 | 37.4±0.1 | -0.16±0.08 | 41.2±0.1 | -0.17±0.09 | 33.4±0.1 | -0.01±0.06 | 31.6±0.1 | 0.02±0.07 |
| | | KF-AQS | 35.6±0.1 | -0.07±0.04 | 37.4±0.1 | -0.15±0.08 | 40.4±0.1 | -0.16±0.09 | 33.1±0.1 | -0.01±0.06 | 31.6±0.1 | 0.02±0.07 |
| | | KF-OMI | 38.7±0.1 | -0.29±0.04 | 41.9±0.1 | -0.42±0.09 | 43.9±0.1 | -0.11±0.09 | 35.6±0.1 | -0.26±0.06 | 33.3±0.1 | -0.41±0.10 |
| | 2005-2009 | KF-AQS | 35.7±0.1 | -0.25±0.14 | 37.7±0.2 | -0.45±0.39 | 40.7±0.1 | -0.97±0.21 | 32.9±0.1 | -0.12±0.35 | 31.5±0.2 | -0.13±0.33 |
| | | KF-OMI | 40.1±0.1 | -0.13±0.18 | 43.5±0.1 | -0.21±0.40 | 43.5±0.1 | -0.70±0.13 | 37.1±0.1 | -0.18±0.48 | 36.5±0.1 | -0.35±0.32 |
| | 2010-2014 | KF-AQS | 36.1±0.1 | -0.51±0.26 | 38.3±0.1 | -0.78±0.62 | 41.3±0.1 | -1.31±0.39 | 33.3±0.1 | -0.17±0.31 | 31.5±0.1 | -0.30±0.65 |
| | | KF-OMI | 39.1±0.1 | -0.89±0.14 | 43.3±0.1 | -1.20±0.56 | 45.1±0.1 | -1.37±0.37 | 35.4±0.1 | -0.41±0.31 | 31.9±0.1 | -0.67±0.63 |
| | 2015-2020 | KF-AQS | 35.1±0.1 | 0.03±0.11 | 36.5±0.1 | -0.05±0.30 | 39.6±0.1 | 0.15±0.35 | 32.9±0.1 | 0.04±0.31 | 31.8±0.1 | 0.09±0.29 |
| | | KF-OMI | 37.1±0.1 | -0.18±0.13 | 39.5±0.1 | -0.43±0.39 | 43.2±0.1 | -0.02±0.28 | 34.4±0.1 | -0.21±0.25 | 31.8±0.1 | -0.03±0.27 |

**Table. 1.** Averages (with units ppb or DU) and trends (with units ppb yr$^{-1}$ or DU yr$^{-1}$) of surface and tropospheric column $O_3$ concentrations in 2005-2020 over the US from observations (AQS and OMI) and a priori and a posteriori (KF) simulations. T1.1): the modeled surface $O_3$ is sampled at the locations and times of AQS surface $O_3$ observations; T1.2): the modeled surface $O_3$ is averaged over the US (land only); T1.3): the output $O_3$ profiles from the a priori and a posteriori simulations are convolved with OMI $O_3$ averaging kernels; T1.4): the output $O_3$ profiles are NOT convolved with OMI $O_3$ averaging kernels. The uncertainties in the averages are calculated using the bootstrapping method. The trends and uncertainties in the trends are calculated using the linear fitting of averages by using the least squares method (see details in the SI).

| Europe | | | Annual | | Spring | | Summer | | Autumn | | Winter | |
|---|---|---|---|---|---|---|---|---|---|---|---|---|
| | | | Mean | Trend | Mean | Trend | Mean | Trend | Mean | Trend | Mean | Trend |
| T2.1 surface (sampled) | 2005-2020 | AirBase | 31.6±0.2 | 0.08±0.04 | 38.5±0.1 | -0.02±0.06 | 40.7±0.2 | 0.01±0.11 | 25.7±0.2 | 0.14±0.05 | 21.4±0.2 | 0.22±0.05 |
| | | a priori | 35.3±0.2 | 0.04±0.03 | 40.3±0.2 | -0.07±0.04 | 46.6±0.2 | -0.07±0.09 | 31.5±0.2 | 0.07±0.05 | 22.9±0.2 | 0.24±0.05 |
| | | KF-AirBase | 32.0±0.1 | 0.05±0.04 | 38.5±0.1 | -0.04±0.06 | 41.3±0.2 | -0.03±0.10 | 26.6±0.2 | 0.09±0.05 | 21.7±0.1 | 0.19±0.04 |
| T2.2 surface | 2005-2020 | a priori | 35.5±0.1 | 0.01±0.02 | 40.3±0.1 | -0.10±0.04 | 46.0±0.2 | -0.09±0.08 | 31.8±0.2 | 0.04±0.04 | 23.9±0.2 | 0.21±0.05 |
| | | KF-AirBase | 32.5±0.1 | 0.01±0.03 | 38.5±0.1 | -0.08±0.04 | 41.1±0.2 | -0.08±0.09 | 27.7±0.1 | 0.04±0.04 | 22.8±0.1 | 0.17±0.04 |
| T2.3 trop. column (convolved) | 2005-2020 | OMI | 36.4±0.1 | -0.15±0.06 | 37.6±0.1 | -0.33±0.14 | 41.0±0.1 | -0.09±0.08 | 34.5±0.1 | -0.12±0.07 | 32.5±0.1 | -0.09±0.11 |
| | | a priori | 32.8±0.1 | -0.09±0.03 | 33.6±0.1 | -0.18±0.06 | 37.3±0.1 | -0.14±0.06 | 31.3±0.1 | -0.03±0.02 | 29.0±0.0 | -0.02±0.05 |
| | | KF-OMI | 35.3±0.1 | -0.25±0.04 | 37.0±0.1 | -0.40±0.09 | 40.5±0.1 | -0.16±0.06 | 33.1±0.1 | -0.22±0.04 | 30.4±0.0 | -0.23±0.05 |
| T2.4 trop. Column | 2005-2020 | a priori | 32.1±0.1 | 0.03±0.03 | 33.7±0.1 | -0.03±0.06 | 37.2±0.1 | 0.06±0.05 | 29.5±0.1 | 0.01±0.04 | 27.9±0.0 | 0.06±0.05 |
| | | KF-AirBase | 31.5±0.1 | 0.03±0.03 | 33.3±0.1 | -0.03±0.06 | 36.2±0.1 | 0.06±0.05 | 28.8±0.1 | 0.01±0.04 | 27.7±0.1 | 0.06±0.05 |
| | | KF-OMI | 35.9±0.1 | -0.36±0.04 | 39.5±0.1 | -0.48±0.07 | 41.4±0.1 | 0.02±0.06 | 32.1±0.1 | -0.38±0.05 | 30.4±0.0 | -0.58±0.11 |
| | 2005-2009 | KF-AirBase | 31.2±0.1 | -0.24±0.08 | 33.1±0.0 | -0.17±0.26 | 35.8±0.1 | -0.39±0.12 | 28.6±0.0 | -0.40±0.21 | 27.3±0.0 | -0.22±0.30 |
| | | KF-OMI | 38.1±0.0 | -0.38±0.22 | 41.6±0.1 | -0.35±0.39 | 40.9±0.1 | -0.06±0.23 | 34.6±0.0 | -0.76±0.33 | 34.9±0.0 | -1.06±0.44 |
| | 2010-2014 | KF-AirBase | 31.4±0.1 | -0.24±0.23 | 33.6±0.0 | -0.58±0.46 | 35.8±0.1 | -0.33±0.32 | 28.7±0.1 | -0.02±0.30 | 27.4±0.0 | -0.16±0.34 |
| | | KF-OMI | 35.7±0.1 | -0.82±0.12 | 40.6±0.1 | -1.30±0.25 | 41.6±0.1 | -0.54±0.33 | 31.5±0.1 | -0.40±0.19 | 28.3±0.0 | -0.69±0.27 |
| | 2015-2020 | KF-AirBase | 31.7±0.1 | 0.03±0.09 | 33.1±0.1 | -0.03±0.20 | 36.8±0.1 | 0.00±0.22 | 28.9±0.1 | 0.09±0.13 | 28.2±0.0 | -0.02±0.20 |
| | | KF-OMI | 34.3±0.1 | -0.26±0.11 | 36.9±0.1 | -0.58±0.14 | 41.6±0.1 | -0.28±0.33 | 30.5±0.1 | -0.19±0.15 | 28.5±0.0 | -0.11±0.20 |

**Table. 2.** Averages (with units ppb or DU) and trends (with units ppb yr[-1] or DU yr[-1]) of surface and tropospheric column $O_3$ concentrations in 2005-2020 over Europe from observations (AirBase and OMI) and a priori and a posteriori (KF) simulations. T2.1): the modeled surface $O_3$ are sampled at the locations and times of AirBase surface $O_3$ observations; T2.2): the modeled surface $O_3$ are averaged over Europe (land only); T2.3): the output $O_3$ profiles from the a priori and a posteriori simulations are convolved with OMI $O_3$ averaging kernels; T2.4): the output $O_3$ profiles are NOT convolved with OMI $O_3$ averaging kernels.

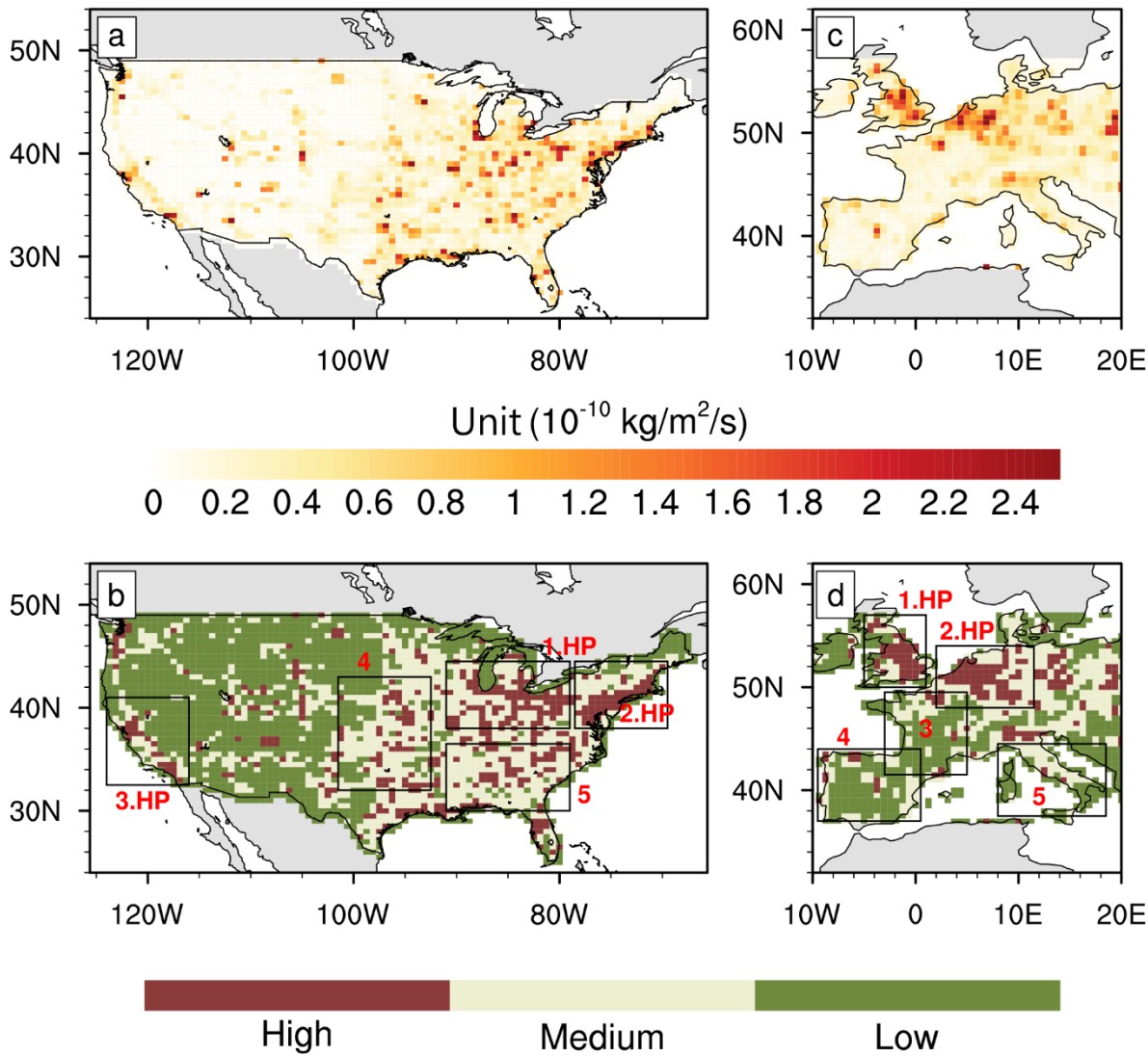

**Fig. 1.** (a) Anthropogenic $NO_x$ emissions over the US in 2015; (b) Region definitions for Great Lakes (#1), Northeast US (#2), West Coast (#3), Middle US (#4) and Southeast US (#5). Regions #1-3 are defined as highly polluted (HP) regions by excluding grids with low and medium anthropogenic $NO_x$ emissions. (c) Anthropogenic $NO_x$ emissions over Europe in 2015; (d) Region definitions for Britain (#1), Central EU (#2), Western EU (#3), Iberian Peninsula (#4) and Apennine Peninsula (#5). Regions #1 and #2 are defined as highly polluted (HP) regions by excluding grids with low and medium anthropogenic $NO_x$ emissions. The different colors (red, gray and green) represent grids with high (highest 15%), medium (15-50%) and low (lowest 50%) anthropogenic $NO_x$ emissions.

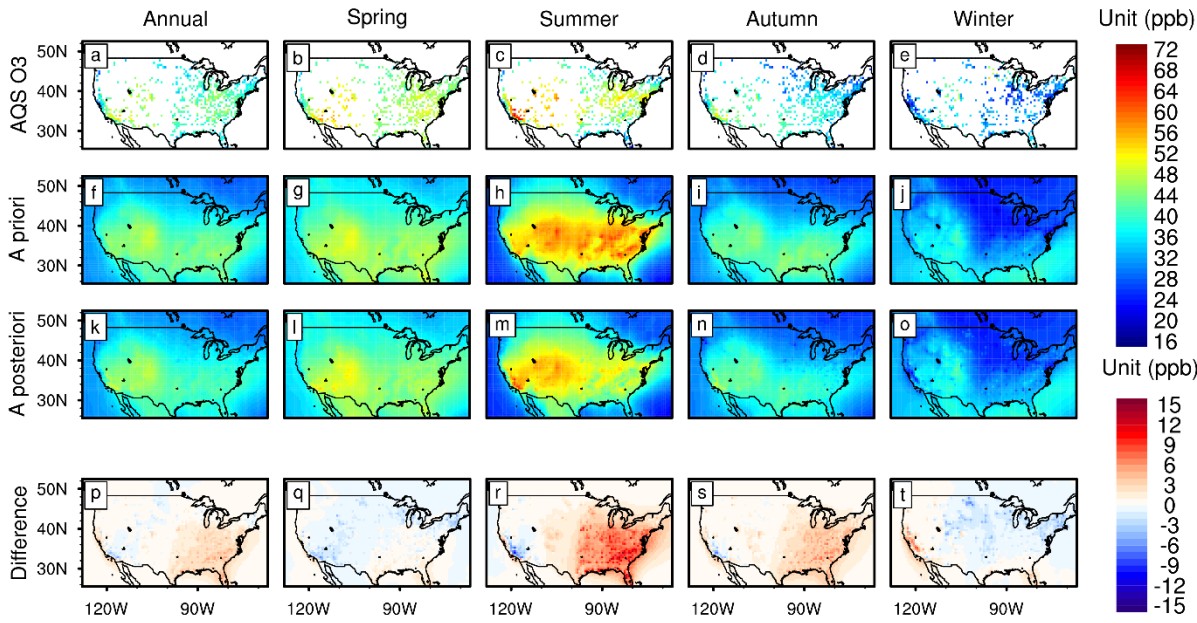

**Fig. 2.** Surface MDA8 $O_3$ over the US in 2005-2020 (annual and seasonal averages) from (a-e) AQS stations; (f-j) GEOS-Chem a priori simulation; (k-o) GEOS-Chem a posteriori simulation by assimilating AQS $O_3$ observations. (p-t) Bias in the a priori simulations calculated by a priori minus a posteriori $O_3$ concentrations.

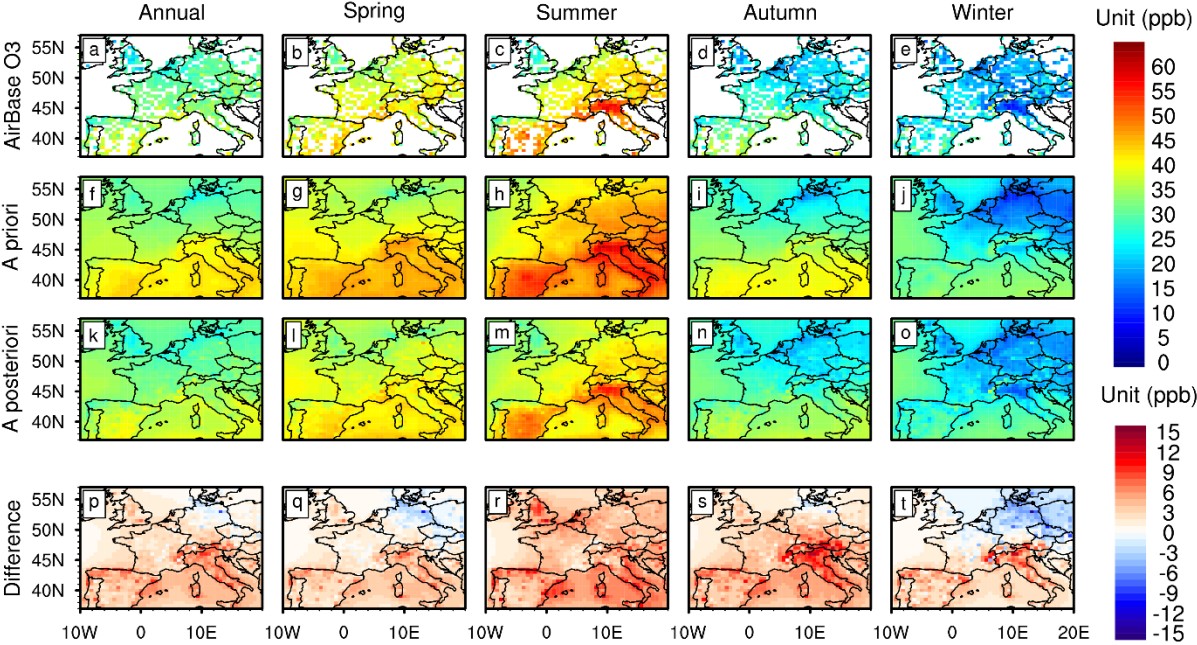

**Fig. 3.** Surface MDA8 $O_3$ over Europe in 2005-2020 (annual and seasonal averages) from (a-e) AirBase stations; (f-j) GEOS-Chem a priori simulation; (k-o) GEOS-Chem a posteriori simulation by assimilating AirBase $O_3$ observations. (p-t) Bias in the a priori simulations calculated by a priori minus a posteriori $O_3$ concentrations.

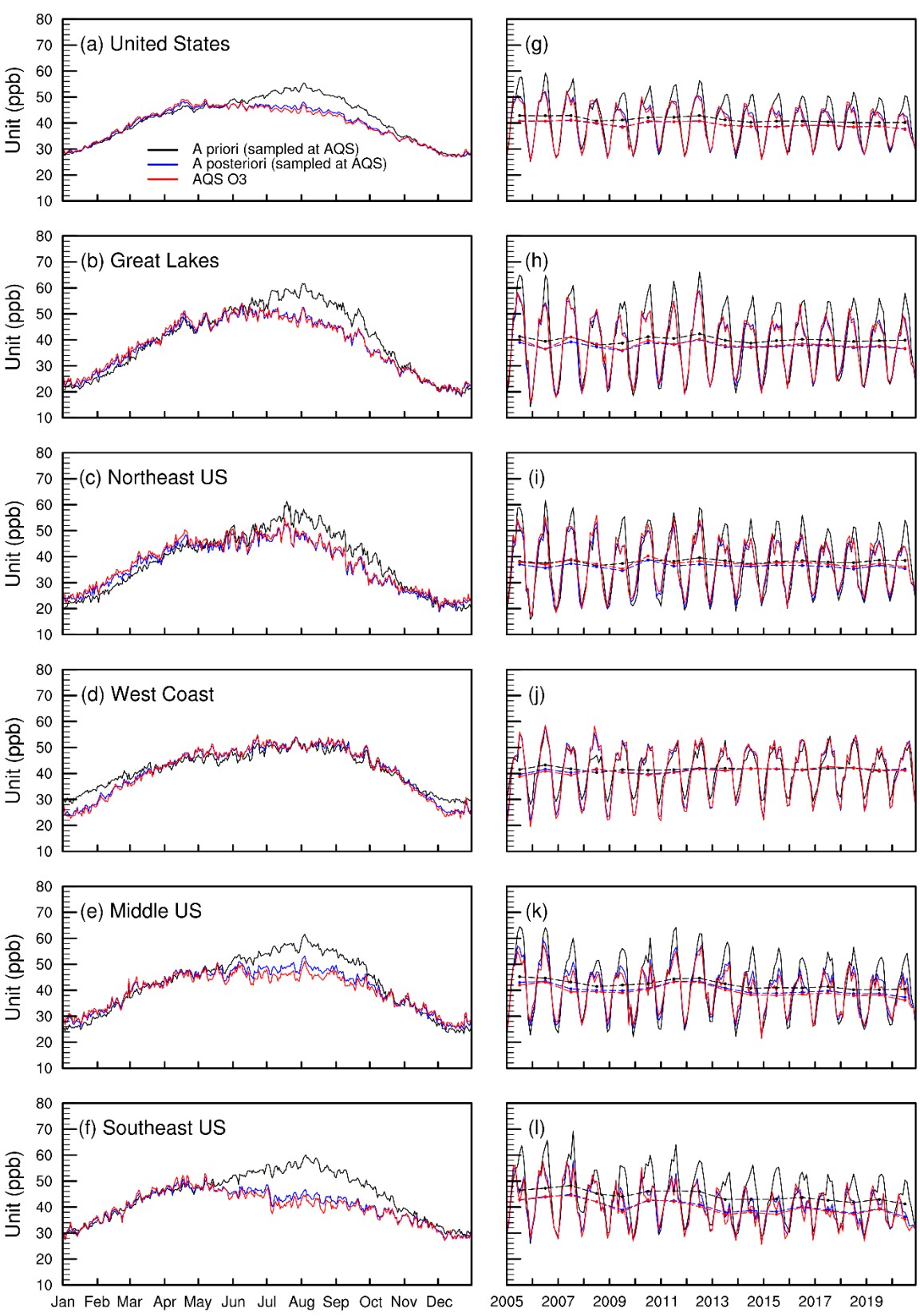

**Fig. 4.** (a-f) Daily averages of surface MDA8 $O_3$ over the US in 2005-2020 from AQS stations (red) and GEOS-Chem a priori (black) and a posteriori (blue) simulations by assimilating AQS $O_3$ observations. (g-l) Monthly averages of MDA8 $O_3$. The dashed lines in panels g-l are annual averages.

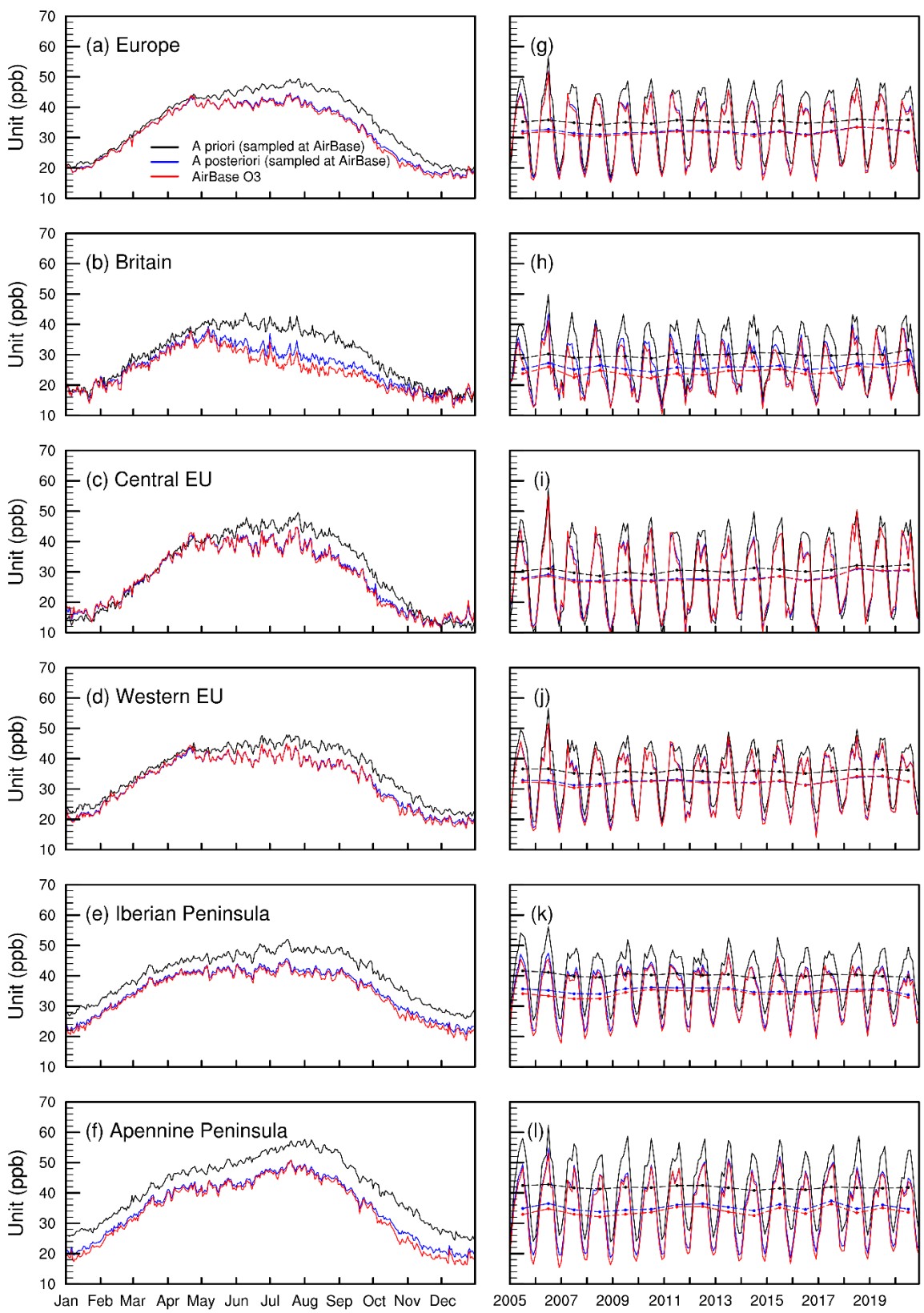

**Fig. 5.** (a-f) Daily averages of surface MDA8 O$_3$ over Europe in 2005-2020 from AirBase stations (red) and GEOS-Chem a priori (black) and a posteriori (blue) simulations by assimilating AirBase O$_3$ observations. (g-l) Monthly averages of MDA8 O$_3$. The dashed lines in panels g-l are annual averages.

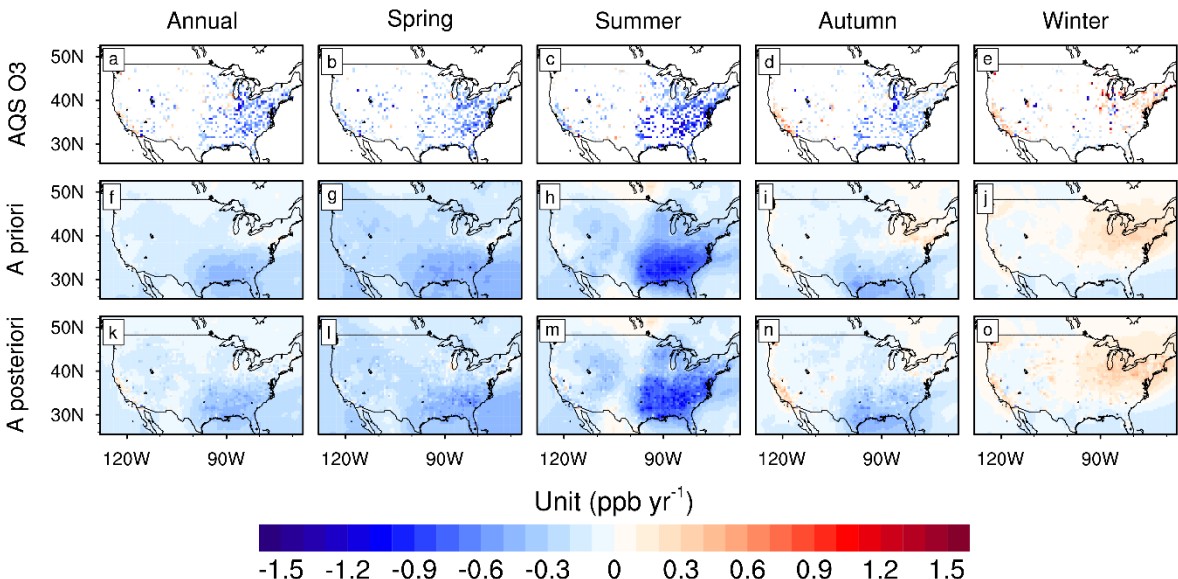

**Fig. 6.** Trends of surface MDA8 $O_3$ over the US in 2005-2020 (annual and seasonal averages) from (a-e) AQS stations; (f-j) GEOS-Chem a priori simulation; (k-o) GEOS-Chem a posteriori simulation by assimilating AQS $O_3$ observations.

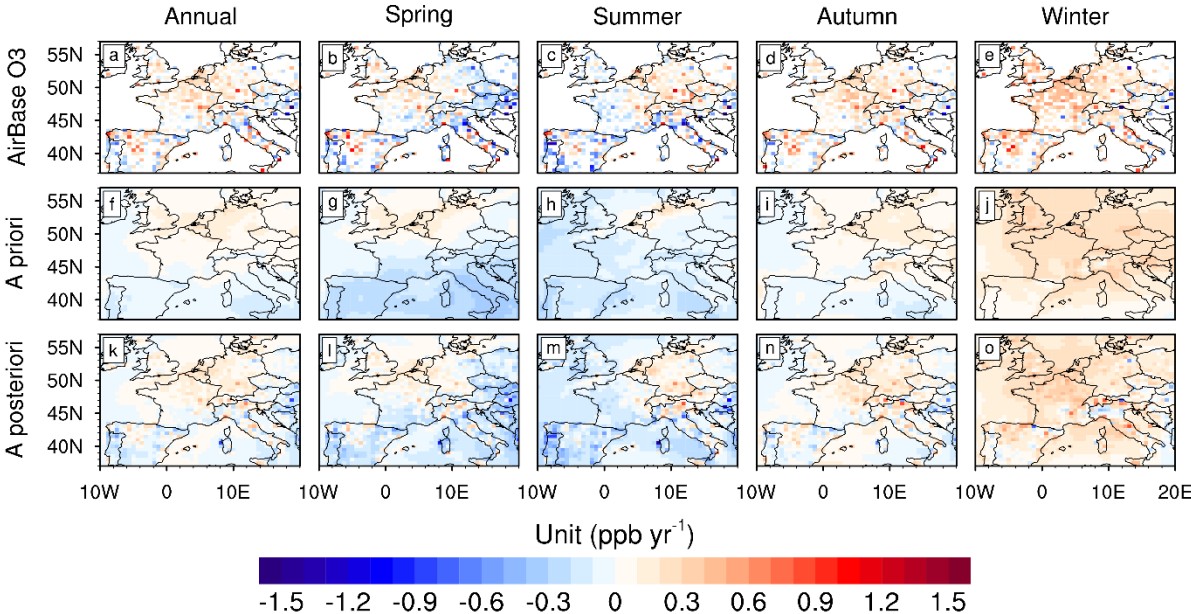

**Fig. 7.** Trends of surface MDA8 $O_3$ over Europe in 2005-2020 (annual and seasonal averages) from (a-e) AirBase stations; (f-j) GEOS-Chem a priori simulation; (k-o) GEOS-Chem a posteriori simulation by assimilating AirBase $O_3$ observations.

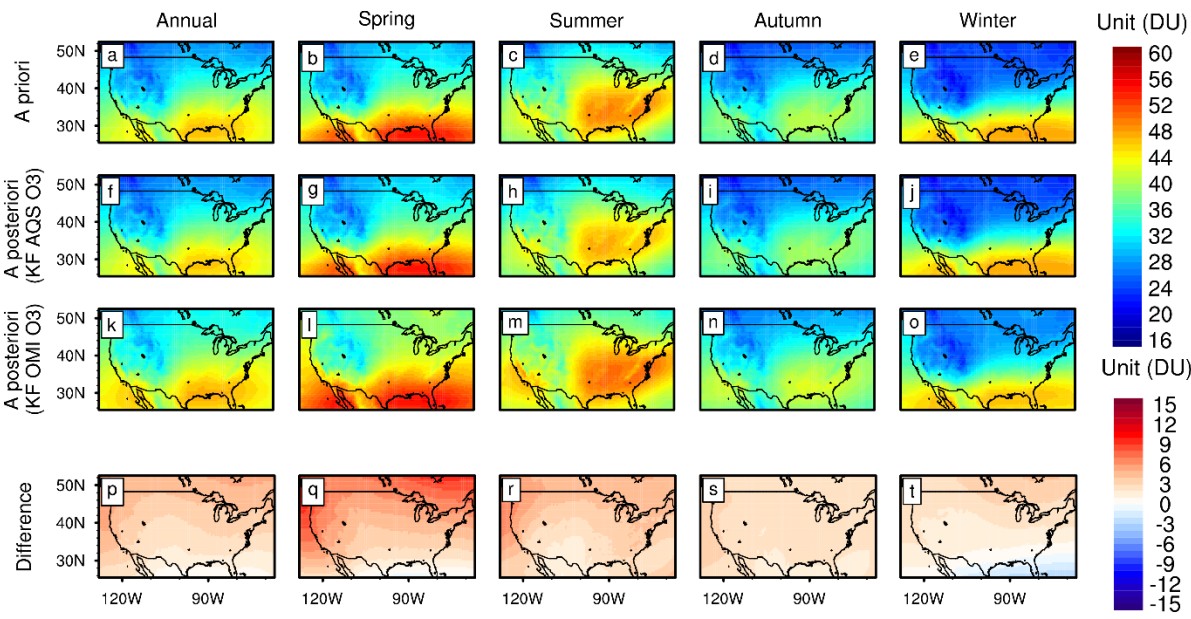

**Fig. 8.** Tropospheric O$_3$ columns over the US in 2005-2020 (annual and seasonal averages) from (a-e) GEOS-Chem a priori simulation; (f-j) Assimilations of AQS surface O$_3$ observations; (k-o) Assimilations of OMI O$_3$ observations. (p-t) Difference in tropospheric O$_3$ columns calculated by OMI-based assimilations minus surface observation-based assimilations.

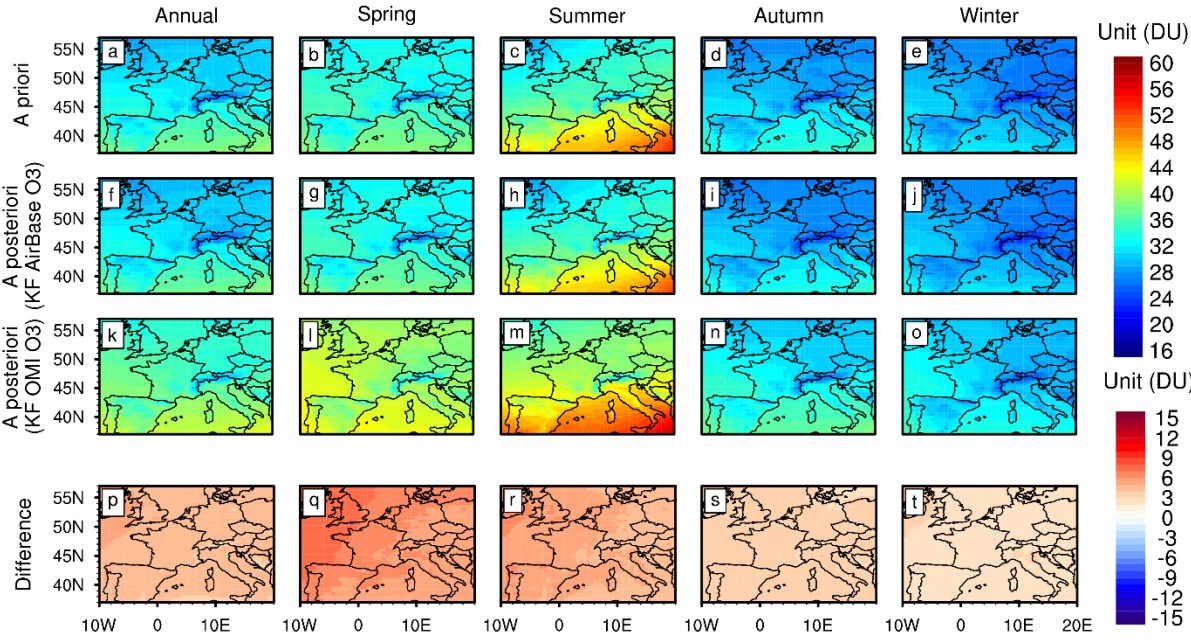

**Fig. 9.** Tropospheric O$_3$ columns over Europe in 2005-2020 (annual and seasonal averages) from (a-e) GEOS-Chem a priori simulation; (f-j) Assimilations of AirBase surface O$_3$ observations; (k-o) Assimilations of OMI O$_3$ observations. (p-t) Difference in tropospheric O$_3$ columns calculated by OMI-based assimilations minus surface observation-based assimilations.

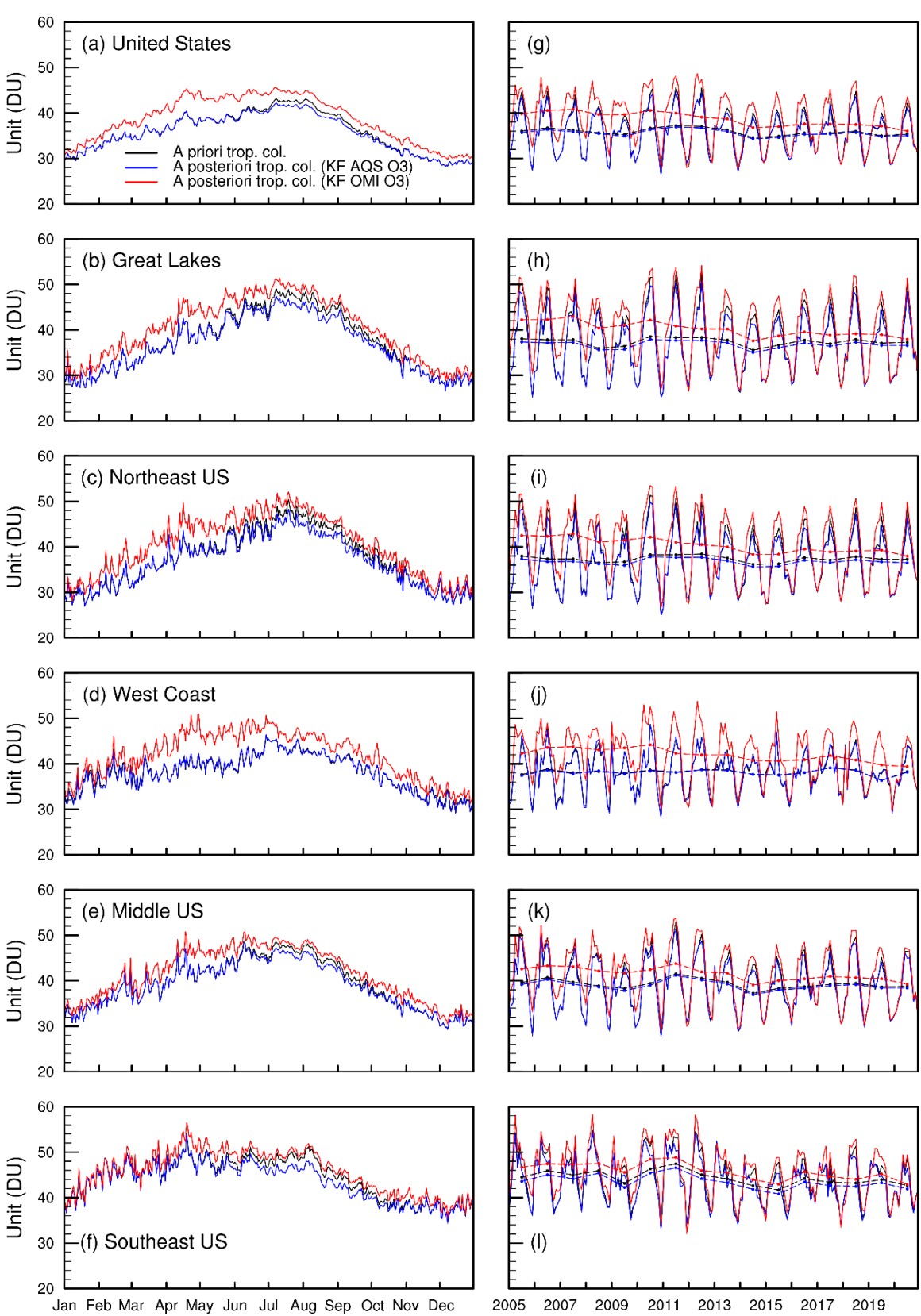

**Fig. 10.** (a-f) Daily averages of tropospheric O$_3$ columns over the US in 2005-2020 from GEOS-Chem a priori simulation (black) and a posteriori simulations by assimilating AQS (blue) and OMI (red) O$_3$ observations. (g-l) Monthly averages of tropospheric O$_3$ columns. The dashed lines in panels g-l are annual averages.

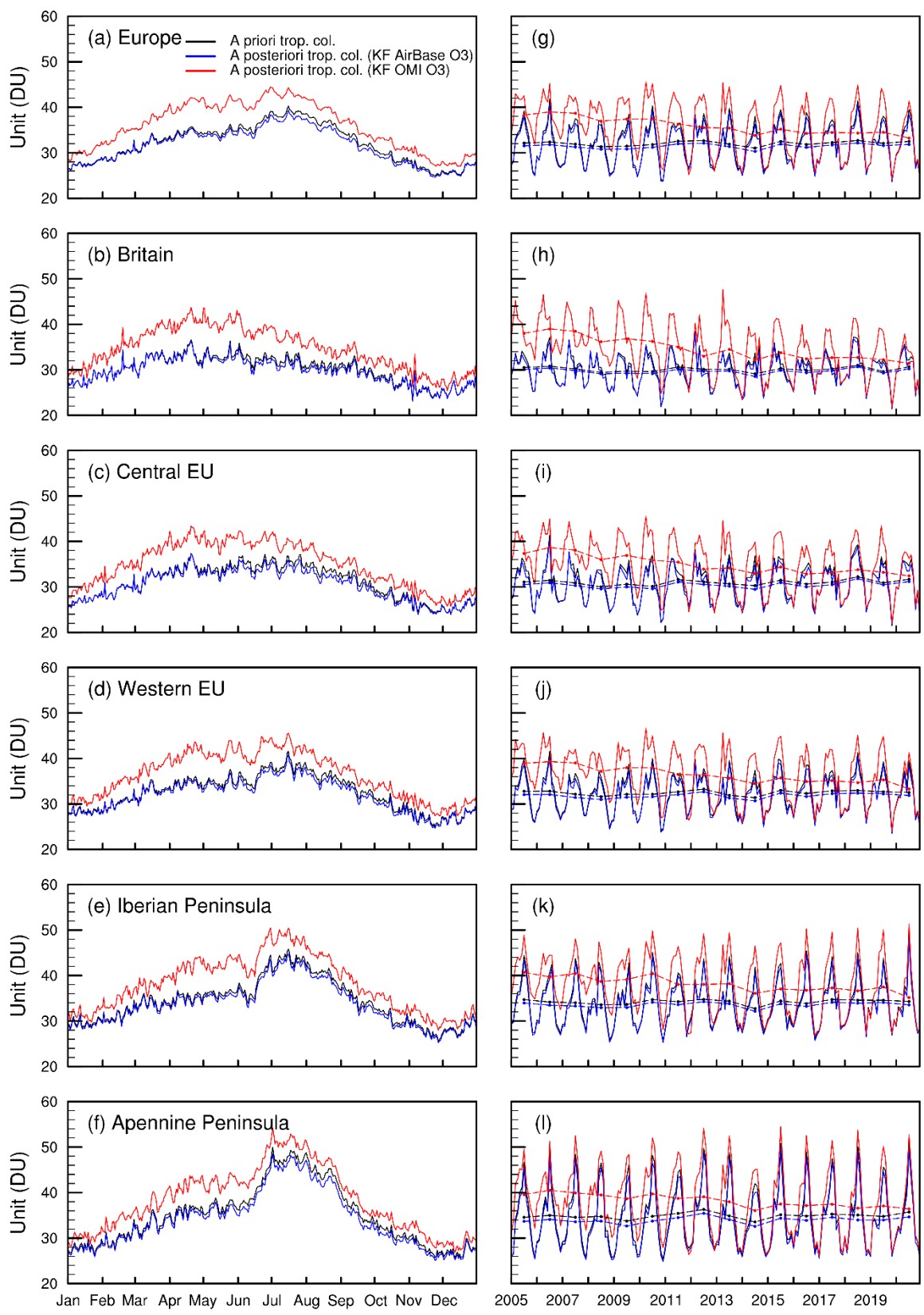

**Fig. 11.** (a-f) Daily averages of tropospheric O₃ columns over Europe in 2005-2020 from GEOS-Chem a priori simulation (black) and a posteriori simulations by assimilating AirBase (blue) and OMI (red) O₃ observations. (g-l) Monthly averages of tropospheric O₃ columns. The dashed lines in panels g-l are annual averages.

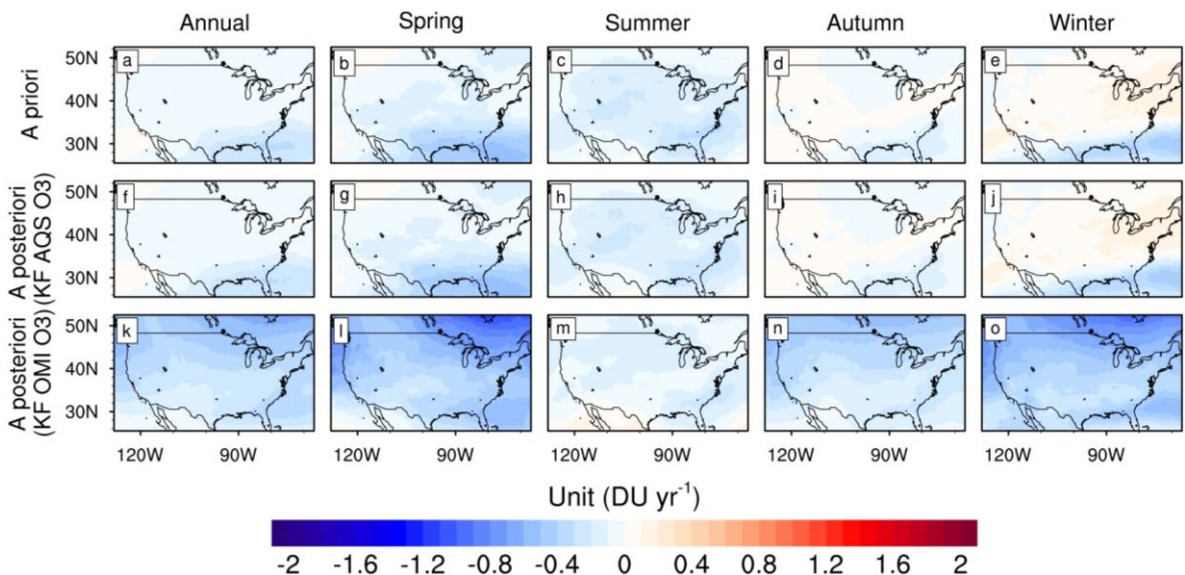

**Fig. 12.** Trends of tropospheric O₃ columns over the US in 2005-2020 (annual and seasonal averages) from (a-e) GEOS-Chem a priori simulation; (f-j) Assimilations of AQS surface O₃ observations; (k-o) Assimilations of OMI O₃ observations.

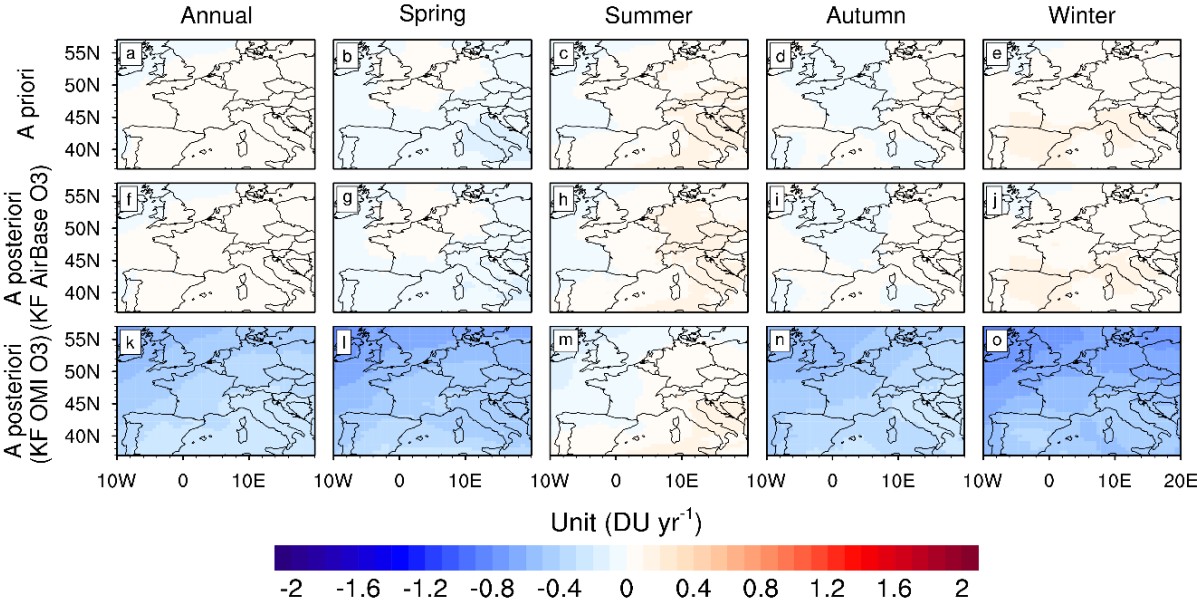

**Fig. 13.** Trends of tropospheric O₃ columns over Europe in 2005-2020 (annual and seasonal averages) from (a-e) GEOS-Chem a priori simulation; (f-j) Assimilations of AirBase surface O₃ observations; (k-o) Assimilations of OMI O₃ observations.