# Peer review of "Rapid O3 assimilations – Part 2: tropospheric O3 changes accompanied by 2 declines in NOx emissions in the US and Europe in 2005-2020 3 Rui Zhu1, Zhaojun Tang1, Xiaokang Chen1, Xiong Liu2 and Zhe Jiang1\"

_Atmospheric Chemistry and Physics, 2023_

## Referee Comment (RC1)

The manuscript titled "Rapid assimilations of O3 observations – Part 2: tropospheric O3 changes in the United States and Europe in 2005-2020" assesses the impact of synoptic patterns on O$_3$ pollution and crop yield in China. This study shows the different trends between surface O3 and tropospheric O3 column, and attempt to explore the underlying driver of trends. The article is well organized. It can be accepted after considering the following suggestions.

Major comments:

The manuscript lacks of the description of methods. This makes it hard for reader to follow the section of results and discussion. I suggest to provide a clear picture of the methods. Also, I did not find the citation of Part 1, Zhu et al. (2023) in Atmos Chem Phys, and it seems that it is under review in Geosci Model Dev.

Some key terms should be explained clearly, such as priori simulations and posteriori simulations and how to assimilate surface O$_3$ and OMI O$_3$ column. In addition, it is important to clarify the difference between assimilation of surface O$_3$ and assimilation of OMI O$_3$ column, and which represents the real O$_3$ column. Otherwise, readers cannot understand clearly.

I suggest to add an analysis about the uncertainties in simulations of O$_3$, observations of OMI O$_3$ column, assimilations of O$_3$ column and O$_3$ trends. It is important to know if the trend of O$_3$ is statistically significant.

Other comments:

Lines 130-133: Another reason for the inverse relationships between surface O$_3$ concentrations and local anthropogenic NOx emissions is the titration of NO on O$_3$. How do you consider the titration effect of NO?

Line 182-185: How do you draw this conclusion since you did not show the trend of local anthropogenic emissions (NO$_X$ and VOC emissions) for different seasons?

Line 209-211: Why does the posteriori simulations of O$_3$ column decreased far faster than the OMI observations? If the OMI observations represent a real O3 column trend, this results exactly suggest that the posteriori simulations of O$_3$ column overestimates the decrease in O$_3$ column. The trends of posteriori simulations of O$_3$ column here (-0.16 DU yr$^{-1}$) differs from that in Section 2.4 (-0.29 DU yr$^{-1}$). Please explains these differences.

Line 26-30: Here you show distinct O3 column trends derived from two methods. You need to clarify which is the real trend, and what we learn from the comparison of two distinct trends.

---

## Author Response (AR1)

We thank the reviewers for their thoughtful and detailed comments. We have revised this manuscript carefully based on the comments. Below we respond to the individual comments.

Reviewer #1

The manuscript titled "Rapid assimilations of O3 observations – Part 2: tropospheric O3 changes in the United States and Europe in 2005-2020" assesses the impact of synoptic patterns on O3 pollution and crop yield in China. This study shows the different trends between surface O3 and tropospheric O3 column, and attempt to explore the underlying driver of trends. The article is well organized. It can be accepted after considering the following suggestions.

**Answer**: Thank you for the comments! The manuscript has been revised based on the comments.

Major comments:

**Question**: The manuscript lacks of the description of methods. This makes it hard for reader to follow the section of results and discussion. I suggest to provide a clear picture of the methods. Also, I did not find the citation of Part 1, Zhu et al. (2023) in Atmos Chem Phys, and it seems that it is under review in Geosci Model Dev.

**Answer**: The manuscript (Part 1) was originally submitted to ACP but was then transferred to GMD because it focuses on the development and performance of the single $O_3$ tracer assimilation system, which matches the scoop of GMD better. As the reviewer suggested, a new Section 2 was included in the revised manuscript to provide descriptions for atmospheric $O_3$ observations and the single $O_3$ tracer simulation and assimilation system used in this work. We are sorry for this confusion!

**Question**: Some key terms should be explained clearly, such as priori simulations and posteriori simulations and how to assimilate surface O3 and OMI O3 column. In addition, it is important to clarify the difference between assimilation of surface O3 and assimilation of OMI O3 column, and which represents the real O3 column. Otherwise, readers cannot understand clearly.

**Answer**: As the answer to the above question, descriptions of atmospheric $O_3$ observations and the single $O_3$ tracer simulation and assimilation system (Section 2) was provided in the revised manuscript.

Furthermore, the difference in the trends of tropospheric $O_3$ columns by assimilating surface and satellite observations is discussed in the revised manuscript: "We note our OMI-based analysis could be affected by the row anomaly issue, although the usage of "row-isolated" data by using across-track positions between 4-11 in this work is expected to reduce the impacts of row anomaly. ... Furthermore, OMI is sensitive to $O_3$ concentrations in the free troposphere; OMI-based assimilations are driven by adjusted regional $O_3$ boundary conditions provided by global OMI $O_3$ assimilations and can reflect optimized adjustments in both local and global background $O_3$ concentrations. In contrast, surface observations are sensitive to local $O_3$ concentrations; surface observation-based assimilations are driven by the a priori $O_3$ boundary conditions,

which thus reflects the optimized adjustments in local contributions and is also affected by lacking optimization on the impacts of $O_3$ precursors due to the single $O_3$ tracer simulations. These factors contributed to the difference in the trends of tropospheric $O_3$ columns by assimilating surface and satellite observations".

Because both OMI and surface observation-based assimilations have their advantages and limitations, we find it is difficult to make a conclusion about which represents the real $O_3$ column better. As discussed in the revised manuscript: "Assimilations of both surface and satellite observations, as shown in this work, are expected to provide more information to better characterization of the changes and uncertainties in free tropospheric $O_3$". It should be noted that the algorithm of the new version of OMI $O_3$ profile retrieval product (v2 in contrast to v0.9.3 in this work) has been finalized recently with improved retrieval accuracy and reduced latitudinal-dependent biases and trend artifacts. We may expect more reliable assimilations of OMI $O_3$ data with the new OMI $O_3$ data in the future.

The manuscript has been revised. Thank the reviewer for pointing out this issue!

**Question**: I suggest to add an analysis about the uncertainties in simulations of O3, observations of OMI O3 column, assimilations of O3 column and O3 trends. It is important to know if the trend of O3 is statistically significant.

**Answer**: Thank the reviewer for this suggestion! Uncertainty estimates have been included in all Tables in the revised manuscript. As described in the revised manuscript: "The uncertainties in the averages are calculated using the bootstrapping method. The trends and uncertainties in the trends are calculated using the linear fitting of averages by using the least squares method (see details in the SI)".

Other comments:

**Question**: Lines 130-133: Another reason for the inverse relationships between surface O3 concentrations and local anthropogenic NOx emissions is the titration of NO on O3. How do you consider the titration effect of NO?

**Answer**: We agree with the reviewer that the titration effect of NO can have an important influence on surface $O_3$ concentrations. However, as a companion paper of Zhu et al. 2023 ($O_3$ changes in China, under review by GMD), here we focus on the comparison in $O_3$ pollution between China and US/Europe. We found the titration effect may not be the dominant role because NOx concentrations are much higher in China while $O_3$ concentrations are still higher in areas with higher $NO_x$ emissions in China. As discussed in the revised manuscript, the discrepancy in $O_3$ pollution between China and US/Europe could be more contributed by the "continuous declines in anthropogenic emissions in the past decades" in the US/Europe and thus, the "impacts of natural sources and meteorological conditions on surface $O_3$ pollution over the US and Europe" is more important than those in China.

Furthermore, the single $O_3$ tracer simulations are driven by archived production (PO3) and loss (LO3) of $O_3$ provided by the full-chemistry simulation, which cannot simulate the nonlinear responses between $O_3$ and its precursors. We expect a limited influence of the single $O_3$ tracer on the assimilations of surface $O_3$ concentrations because of the

continuous assimilations of surface O₃ observations with hourly temporal resolutions. However, it may have impacts on free tropospheric O₃. As discussed in the revised manuscript: "surface observation-based assimilations … is also affected by lacking optimization on the impacts of O₃ precursors due to the single O₃ tracer simulations".

**Question**: Line 182-185: How do you draw this conclusion since you did not show the trend of local anthropogenic emissions (NOX and VOC emissions) for different seasons?

**Answer**: This sentence has been removed. Thank the reviewer for pointing out this issue!

**Question**: Line 209-211: Why does the posteriori simulations of O3 column decreased far faster than the OMI observations? If the OMI observations represent a real O3 column trend, this results exactly suggest that the posteriori simulations of O3 column overestimates the decrease in O3 column. The trends of posteriori simulations of O3 column here (-0.16 DU yr-1) differs from that in Section 2.4 (-0.29 DU yr-1). Please explains these differences.

**Answer**: The deviations in the trends between assimilations and OMI observations are caused by the adjustments to regional O₃ boundary conditions in the nested assimilations by assimilating global OMI O₃ observations. It reflects the different changes in OMI O₃ between US/Europe continents and global backgrounds and may not be interpreted as overestimations in the decreasing trends.

As discussed in the revised manuscript: "the mean tropospheric O₃ columns over the US in 2005 are 36.5 DU in OMI observations, and 35.9 and 37.5 DU in the assimilations by reading a priori and adjusted O₃ boundary conditions, respectively; the mean tropospheric O₃ columns over Europe in 2005 are 37.5 DU in OMI observations, and 34.6 and 36.9 DU in the assimilations by reading a priori and adjusted O₃ boundary conditions, respectively". The difference due to the usage of different regional O₃ boundary conditions, i.e., approximately 2 DU in 2005, is large enough to lead to small deviations in the trends, e.g., -0.16 and -0.01 DU yr$^{-1}$ over the US.

The different trends in tropospheric O₃ columns, i.e., -0.16 DU yr$^{-1}$ in Section 3.3 and -0.29 DU yr$^{-1}$ in Section 3.4 are caused by the effect of convolution of OMI averaging kernels. The averaging kernels are not applied in the tropospheric O₃ columns in the later part of Section 3.3 as well as Section 3.4.

**Question**: Line 26-30: Here you show distinct O3 column trends derived from two methods. You need to clarify which is the real trend, and what we learn from the comparison of two distinct trends.

**Answer**: As the answer to the above question, both assimilations of satellite and surface O₃ observations have their advantages and limitations, it is thus difficult to make a conclusion about which one is more reliable. The following sentence was added in the Abstract to clarify this point: "The discrepancy in assimilated tropospheric O₃ columns further indicates the possible uncertainties in the derived tropospheric O₃ changes".

Reviewer #2

In this study, the authors applied the newly developed tagged-O3 model to investigate the tropospheric and surface NO2- and O3 changes in both US and Europe. To improve the accuracy of the long-term changes, the authors also applied both the surface observation assimilation and satellite column data assimilation. I appreciate the efforts the authors devoted to improve the representation of long-term ozone changes in both surface and tropospheric by combing multiple data sources. However, the authors need to make necessary adjustments before considering to be published in the journal.

**Answer**: Thank you for the comments! The manuscript has been revised based on the comments.

**Question**: Add a new section for the methodology. Briefly discuss the tagged-O3 mode of the GEOS-Chem model in Zhu et al., 2023a GMD paper. Otherwise, the manuscript will not be complete. Also, describe how the simulations were performed. The authors referred to the "simulation" results multiple times in the manuscript, but there were no descriptions how the simulations were performed. Such as line 21.

**Answer**: A new Section 2 was included in the revised manuscript to provide descriptions for atmospheric $O_3$ observations and the single $O_3$ tracer simulation and assimilation system used in this work. Thank the reviewer for this suggestion!

**Question**: Abstract: line 29-30: I did not quite get how the slowed declines of tropospheric NO2 was related with the tropospheric O3 in 2010-2014? Please explain. Also, please explain how should we comprehend the different tropospheric O3 columns trends by using the surface observation-based assimilations and OMI-based assimilations.

**Answer**: The decline in OMI-based tropospheric $NO_2$ columns mainly happened in 2005-2010, and the decreasing trends have slowed since 2010. In contrast, the OMI-based decreases in tropospheric $O_3$ mainly occurred in 2010-2014. We can thus conclude that "Our analysis thus suggests limited impacts of local emission declines on tropospheric $O_3$ over the US and Europe" because the rapid declines in tropospheric $NO_2$ columns in 2005-2010 corresponds to relatively flat trends in tropospheric $O_3$. The Abstract has been revised to clarify "the reported slowed declines in free tropospheric $NO_2$ since 2010". The discussion in the Conclusion Section was also revised to clarify this point.

Furthermore, as clarified in the revised version, both OMI and surface observation-based assimilations have their advantages and limitations: "We note our OMI-based analysis could be affected by the row anomaly issue, although the usage of "row-isolated" data by using across-track positions between 4-11 in this work is expected to reduce the impacts of row anomaly. ... Furthermore, OMI is sensitive to $O_3$ concentrations in the free troposphere; OMI-based assimilations are driven by adjusted regional $O_3$ boundary conditions provided by global OMI $O_3$ assimilations and can reflect optimized adjustments in both local and global background $O_3$ concentrations. In contrast, surface observations are sensitive to local $O_3$ concentrations; surface observation-based assimilations are driven by the a priori $O_3$ boundary conditions, which thus reflects the optimized adjustments in local contributions and is also affected by lacking optimization on the impacts of $O_3$ precursors due to the single $O_3$ tracer

simulations. These factors contributed to the difference in the trends of tropospheric $O_3$ columns by assimilating surface and satellite observations".

Consequently, it is difficult to make a conclusion about which represents the real $O_3$ column better. As discussed in the revised manuscript: "Assimilations of both surface and satellite observations, as shown in this work, are expected to provide more information to better characterization of the changes and uncertainties in free tropospheric $O_3$". The following sentence was added in the Abstract to clarify this point: "The discrepancy in assimilated tropospheric $O_3$ columns further indicates the possible uncertainties in the derived tropospheric $O_3$ changes".

Editorial comments:

**Question**: All the figures are referred with captive letters in the main manuscript, while in the plots, it is in lower case.

**Answer**: Changed.

**Question**: Suggest to change Zhu et al., 2023 to Zhu et al., 2023a.

**Answer**: The manuscript (Part 1) was originally submitted to ACP but was then transferred to GMD. The reference information has been updated to avoid possible confusion.

**Question**: Line 137-138: I did not quite get the meaning here. What does the authors mean by referring "Following Jiang et al., 2022". Did the authors manually reduce their emissions by 53% or 19% for US and Europe in their simulation? However, my instinct understanding will be that the emissions in the US and Europe declined that percentage from 2005 to 2020. Please explain.

**Answer**: GEOS-Chem model (v12-8-1) uses NEI2011 and CEDS for anthropogenic emissions for the US and Europe, respectively. The reference year for NEI2011 inventory is 2011 with annual scaling factors in 2006-2013; the reference year for CEDS inventory is 2010 with annual scaling factors in 2005-2014. Consequently, as the reviewer indicated, the annual total emissions must be adjusted manually in the simulations using additional annual scaling factors.

As clarified in the revised manuscript: "Following Jiang et al. (2022), the total anthropogenic $NO_x$ and VOC emissions in the GEOS-Chem model are scaled with the corresponding bottom-up inventories (NEI2014 for the US and ECLIPSE for Europe) so that the modeled surface nitrogen dioxide ($NO_2$) and VOC concentrations in the a priori simulations are identical to Jiang et al. (2022) in 2005-2018. The total anthropogenic $NO_x$ and VOC emissions in 2019-2020 in the US and Europe are further scaled based on linear projections. The total anthropogenic $NO_x$ emissions in the a priori simulations declined by 53% (US) and 50% (Europe) in 2005-2020. The total anthropogenic VOC emissions in the a priori simulations declined by 19% (US) and 33% (Europe) in 2005-2020".

Figures

**Question**: I understand the authors have already provided more than enough figures in this manuscript. However I feel it is a little confusing by putting US and Europe together with

the same letters for each country, such as Fig. 2, Fig 5. Maybe the authors can separate the US and Europe by adding the country/region names on top of each figure?

**Answer**: Thank the reviewer for this suggestion! The figures showing both the US and Europe have been split into individual figures. To keep a reasonable number of total figures, some less important figures were moved to the SI in the revised manuscript.